# Root Cause Analysis of Anomalies in Multivariate Time Series through Granger Causal Discovery

**Xiao Han**[1]    **Saima Absar**[2]    **Lu Zhang**[2]    **Shuhan Yuan**[1]
[1]Utah State University, [2]University of Arkansas
{xiao.han,shuhan.yuan}@usu.edu, {sa059,lz006}@uark.edu

## Abstract

Identifying the root causes of anomalies in multivariate time series is challenging due to the complex dependencies among the series. In this paper, we propose a comprehensive approach called AERCA that inherently integrates Granger causal discovery with root cause analysis. By defining anomalies as interventions on the exogenous variables of time series, AERCA not only learns the Granger causality among time series but also explicitly models the distributions of exogenous variables under normal conditions. AERCA then identifies the root causes of anomalies by highlighting exogenous variables that significantly deviate from their normal states. Experiments on multiple synthetic and real-world datasets demonstrate that AERCA can accurately capture the causal relationships among time series and effectively identify the root causes of anomalies.

## 1 Introduction

Root cause analysis on multivariate time series data, which is to identify the underlying causes of an anomaly, has a wide spectrum of applications in various domains, such as diagnosing the fault of online cloud-based systems or cyber-physical systems (Jayathilaka et al., 2017; Jeyakumar et al., 2019; Soldani & Brogi, 2022; Yu et al., 2021). Traditional approaches, which manually trace the root cause based on the topology of the systems, have become impractical due to increasing system complexity, leading to a greater focus on data-driven methods. One promising direction is based on a causal framework, which models system components and their dependencies via a causal graph and then traces how the failure of one component might propagate through the system (Assaad et al., 2023a; Li et al., 2022; Zhang et al., 2021; Ikram et al., 2022; Wang et al., 2023b; Okati et al., 2024). For instance, for a cyber-physical system like a water treatment plant equipped with multiple sensors—such as water level, pH level, and electrical conductivity—that generate multivariate time series data, if an attacker overdoses sodium hydroxide, it could lead to abnormal readings in metrics like pH level and electrical conductivity. Root cause analysis aims to identify the root cause of such abnormal behavior, even when the time series data create ripple effects across other metrics—for example, an increase in sodium hydroxide leading to abnormalities in additional measurements.

Despite the advantages of providing a scalable and systematic way of understanding the relationships and causal chains in complex systems, existing causal inference-based root cause analysis approaches usually suffer from various limitations. For example, Budhathoki et al. (2022) and Assaad et al. (2023b) assume the causal relationships as prior knowledge, which may not be feasible in real cases. On the other hand, although some approaches (Yang et al., 2022; Meng et al., 2020; Wang et al., 2018b) try to learn the causal structures from the observational data, they usually leverage the existing causal discovery algorithms, which do not consider the need for identifying root causes.

In this paper, we propose a comprehensive approach that inherently integrates Granger causal discovery with root cause analysis. We treat the root cause of the anomaly, such as an overdose of sodium hydroxide, as an intervention on the exogenous variables in a structural causal model (SCM). We refer to this as the exogenous intervention, where the exogenous variables follow a stable distri-

bution under normal conditions but undergo interventions when anomalies occur[1]. Under this core idea, we identify the key to root cause identification, which is to model the normality of exogenous variables for multivariate time series and then highlight abnormal exogenous variables. The current causal discovery approaches mainly focus on identifying the causal structures among time series/endogenous variables without explicitly modeling the impact of exogenous variables, making them unsuitable for locating the root cause of an anomaly due to exogenous interventions. Therefore, to achieve our goal, we propose a novel autoencoder-based framework for root cause analysis, referred to as AERCA. This framework identifies Granger causal relationships in time series by explicitly modeling the distributions of exogenous variables, which serves as the foundation for our root cause localization approach.

Specifically, to model the data generation process, i.e., the causal relationships as well as the distributions of exogenous variables, the encoder models the abductive reasoning process to derive the exogenous variable for each time series. Based on our core assumption that the exogenous variables are mutually independent, we establish effective constraints to ensure this independence. Meanwhile, the decoder learns a deductive reasoning process to infer the observed data from the exogenous variables. We theoretically show that to predict the input at time $t$, rather than using exogenous variables of all time steps before $t$, the decoder only needs to take in the exogenous variables and observed time series from a window prior to $t$. We train AERCA on the normal data. Then, upon deployment, if the encoder-derived values of exogenous variables significantly deviate from the norm, the corresponding time series are highly likely to be the root cause of the anomaly.

The contributions of this paper are as follows: 1) we propose a novel encoder-decoder structure for Granger causal discovery, which can not only learn the causal relationships between time series but also capture the distribution of exogenous variables; 2) based on the learned structural causal model, AERCA can not only identify the root cause time series but also highlight the root cause time steps; 3) experimental results on multiple datasets show that AERCA can achieve state-of-the-art performance on both Granger causal discovery and root cause identification.

## 2 RELATED WORK

Understanding the root cause of an anomaly has received increasing attention because of wide real-world applications. Accurate root cause localization can help domain users understand and mitigate abnormal behaviors.

The mainstream approaches in root cause analysis follow a two-step framework: identifying the dependency between variables from observational data and then localizing the root cause by exploring the dependency graph. Therefore, the key step is to build the dependency graph. Traditionally, domain knowledge or a systems tool can be leveraged to build the dependency graph. For example, in a microservice system, a directed edge between two nodes usually indicates a system call (Kim et al., 2013; Weng et al., 2018; Wang et al., 2018a; Yu et al., 2021).

However, as the system becomes sophisticated, it becomes impractical to build the dependency graph based on domain knowledge, and the call graph learned by system tools may not represent the true dependency between sensors (Kim et al., 2013). Therefore, data-driven approaches are now commonly used for learning the dependency between variables. For example, various deep neural networks are developed to capture the temporal and spatial correlations in the multivariate time series for root cause analysis (Zhang et al., 2019; Tuli et al., 2022; Zhao et al., 2020).

Recently, causal inference-based root cause analysis has received increasing attention, which models the anomaly as data under intervention (Assaad et al., 2023a; Li et al., 2022). Under this assumption, root cause localization is to identify the intervention on observational data (Li et al., 2022). Several approaches leverage the PC algorithm (Spirtes et al., 2001) or its variance to build the causal graph by using the conditional independent test (Zhang et al., 2021; Ikram et al., 2022). Some approaches also leverage the graph neural networks to learn the causal relationships between nodes by simulating the data generation process (Wang et al., 2023b;a).

---

[1]Note that not all attacks can be treated as interventions on exogenous variables. Understanding the nature of anomalies is crucial before applying our method to real-world applications.

In this work, we propose a comprehensive approach that inherently integrates Granger causal discovery with root cause analysis. By assuming that anomalies are caused by exogenous interventions, we introduce a novel method for Granger causal discovery that explicitly models the distribution of exogenous variables. Consequently, unlike existing studies that can only locate the root-cause time series without specifying the abnormal time steps, our approach identifies the root cause as the time series receiving exogenous interventions at specific time steps, providing much more informative and precise localization.

## 3  PRELIMINARY: GRANGER CAUSALITY

Granger causality (Granger, 1969; Dahlhaus & Eichler, 2003) is commonly used for modeling causal relationships in multivariate time series. The key assumption is that if the prediction of the future value $Y$ can be improved by knowing past elements of $X$, then $X$ "Granger causes" $Y$. Granger causality was originally defined for linear relationships, while recently, the non-linear Granger causality has been proposed (Tank et al., 2021; Assaad et al., 2022):

Let a stationary time-series as $\mathbf{X} = (\mathbf{x}_1, \ldots, \mathbf{x}_t, \ldots, \mathbf{x}_T)$, where $\mathbf{x}_t \in \mathbb{R}^d$ is a d-dimensional vector (e.g., d-dimensional time series data from $d$ sensors) at a specific time $t$. Suppose that the true data generation mechanism is defined in the form of

$$x_t^{(j)} := f^{(j)}(\mathbf{x}_{\leq t-1}^{(1)}, \cdots, \mathbf{x}_{\leq t-1}^{(d)}) + u_t^{(j)}, \text{ for } 1 \leq j \leq d, \tag{1}$$

where $\mathbf{x}_{\leq t-1}^{(j)} = [\cdots, x_{t-2}^{(j)}, x_{t-1}^{(j)}]$ denotes the past of series $j$; $u_t^{(j)} \in \mathbf{u}^{(j)}$ indicates exogenous variable for time series $j$ at time step $t$; $f^{(j)}(\cdot)$ is a function for time series $j$ that captures how the past values impact the future values of $\mathbf{x}^{(j)}$. The time series $i$ Granger causes $j$, if $f^{(j)}$ depends on $\mathbf{x}_{\leq t-1}^{(i)}$, i.e., $\exists \mathbf{x}'^{(i)}_{\leq t-1} \neq \mathbf{x}_{\leq t-1}^{(i)} : f^{(j)}(\mathbf{x}_{\leq t-1}^{(1)}, \cdots, \mathbf{x}'^{(i)}_{\leq t-1}, \cdots, \mathbf{x}_{\leq t-1}^{(d)}) \neq f^{(j)}(\mathbf{x}_{\leq t-1}^{(1)}, \cdots, \mathbf{x}_{\leq t-1}^{(i)}, \cdots, \mathbf{x}_{\leq t-1}^{(d)})$. (Tank et al., 2021; Marcinkevičs & Vogt, 2021; Shojaie & Fox, 2022)

**Limitations of Granger Causality**. While Granger causality is a valuable method for detecting temporal causal dependencies, it is important to understand its limitations. Specifically, Granger causality assumes no hidden confounding, i.e., all relevant variables influencing the causal relationship are observed and included in the model, and no instantaneous effects between variables, i.e., the influence of one variable on another is not immediate but occurs with some time lag. Violating these assumptions can lead to erroneous conclusions in Granger causality analysis, highlighting the importance of careful assessment of assumptions and consideration of alternative models.

## 4  METHODOLOGY

### 4.1  PROBLEM FORMULATION AND FRAMEWORK

Based on the structural equation of multivariate time series defined in Eq. 1, in this work, we focus on the anomaly $\tilde{x}_t^{(j)}$ caused by exogenous interventions on a single or multiple time series, leading to a significantly deviating value in its exogenous variable $\hat{u}_t^{(j)}$, which can be defined as

$$\tilde{x}_t^{(j)} = f^{(j)}(\mathbf{x}_{\leq t-1}^{(1)}, \cdots, \mathbf{x}_{\leq t-1}^{(d)}) + \hat{u}_t^{(j)} = f^{(j)}(\mathbf{x}_{\leq t-1}^{(1)}, \cdots, \mathbf{x}_{\leq t-1}^{(d)}) + u_t^{(j)} + \epsilon_t^{(j)}, \text{ for } 1 \leq j \leq d, \tag{2}$$

where $\hat{u}_t^{(j)} = u_t^{(j)} + \epsilon_t^{(j)}$ with an anomaly term $\epsilon_t^{(j)}$. Note that the abnormal time series caused by exogenous interventions can be either a point anomaly or a sequential anomaly. The point anomaly can be due to an exogenous intervention on a specific time series at a time step. In contrast, a sequence anomaly can be caused by the propagation of an exogenous intervention through time by following the causal structural model or a consistent exogenous intervention over time steps.

Therefore, an informative root cause analysis shows not just the time series but also the time steps receiving the exogenous intervention. Based on this motivation, we define the task of root cause identification below.

**Definition 1.** *The root cause identification is to locate the time series/variables $(j)$ at specific time step(s) $t$ with the abnormal exogenous variable $\hat{u}_t^{(j)}$.*

For the anomaly caused by the exogenous interventions, to achieve the root cause analysis, we learn the Granger causality in multivariate time series by explicitly modeling the distribution of exogenous variables. To this end, we develop an encoder-decoder structure for root cause analysis, called AERCA, which can calculate the exogenous variable for each time series at a specific time step. AERCA explicitly computes the exogenous variables via an encoder, and a decoder predicts the current value by simulating the data generation mechanism defined by the Granger causality. By training the encoder-decoder structure on the normal time series, the model can capture the distribution of exogenous variables in the normal status. When an exogenous intervention occurs, the derived exogenous variables should significantly differ from the normal ones. Meanwhile, because we explicitly derive the exogenous variables at each time step, even if the time series is still abnormal due to the error propagation through time, AERCA can distinguish the root cause from the downstream impact. Figure 1 shows the framework of AERCA. In the following, we explain each components of the framework.

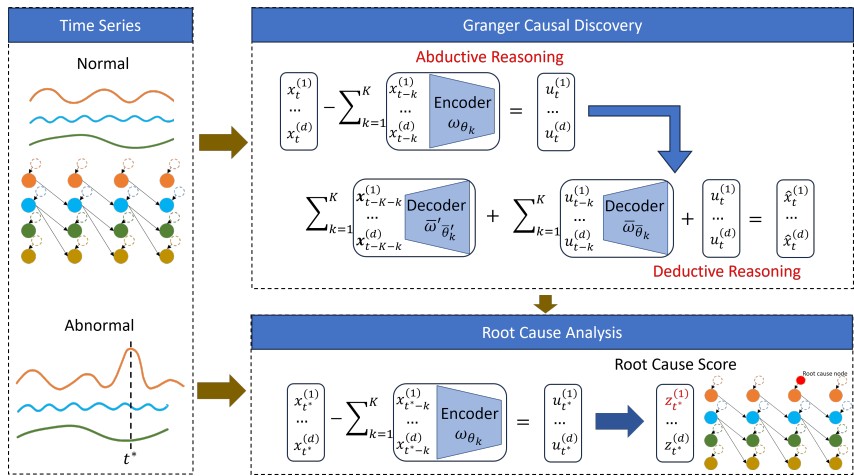

Figure 1: The overview of AERCA.

## 4.2 GRANGER CAUSAL DISCOVERY

**Motivation**. To model the data generation process, i.e., the causal relationships as well as the distributions of exogenous variables, we adopt the encoder-decoder structure to simulate both the abductive and deductive reasoning processes. Abductive reasoning is to seek the most plausible explanations, i.e., to infer the most likely exogenous variables (causes) that could have generated the observed time series data. As shown in Eq. 1, based on the Granger causality, the value of the time series at step $t$ is a function of past time series plus an exogenous term at the current step, i.e., $\mathbf{x}_t := f(\mathbf{x}_{\leq t-1}) + \mathbf{u}_t$, with simplified notations. To simulate abductive reasoning, the encoder derives the exogenous variables based on the observed data by rewriting Eq. 1 as

$$\mathbf{u}_t := \mathbf{x}_t - f(\mathbf{x}_{\leq t-1}). \tag{3}$$

On the other hand, deductive reasoning derives effects from known causes, i.e., reconstructing the observed data from exogenous variables. By recursively resolving each previous time step—such as expressing $\mathbf{x}_{t-1}$ in terms of its predecessor $\mathbf{x}_{t-2}$, and continuing this process backward to the first time step—we can rewrite Eq. 1 in a different way as a function of the exogenous variables:

$$\mathbf{x}_t = \tilde{f}(\mathbf{u}_{\leq t-1}) + \mathbf{u}_t, \tag{4}$$

which shows that the observed data at step $t$ is represented as a function $\tilde{f}(\cdot)$ of all preceding exogenous variables. Within the encoder-decoder framework, this function acts as the decoder for reconstructing the observed data directly from the exogenous variables.

Based on the above analysis, we develop an encoder-decoder structure, where the encoder learns Granger causal relationships $f(\cdot)$ by using past time series values as input to compute the exogenous

variables, simulating Eq. 3. The decoder $\tilde{f}(\cdot)$ then takes these exogenous variables from the encoder as input to reconstruct the value of the current time step $\mathbf{x}_t$, simulating Eq. 4.

**Encoder-decorder Structure**. Given normal multivariate time series $\mathbf{X} = (\mathbf{x}_1, \ldots, \mathbf{x}_t, \ldots, \mathbf{x}_T)$, we define a window with length $K$ as $\mathbf{W}_t = (\mathbf{x}_{t-K+1}, ..., \mathbf{x}_t)$ and convert a time series $\mathbf{X}$ to a sequence of sliding windows $\mathcal{W} = (\mathbf{W}_K, \mathbf{W}_{K+1}, ..., \mathbf{W}_T)$. We first aim to learn the Granger causality of time series in a window, i.e., a window causal graph (Assaad et al., 2022).

Given a time series window, we first parameterize the Granger causality in time series defined in Eq. 1 as

$$\mathbf{x}_t = \sum_{k=1}^{K} \omega_{\theta_k}(\mathbf{x}_{t-k})\mathbf{x}_{t-k} + \mathbf{u}_t, \tag{5}$$

where $\omega_{\theta_k}(\mathbf{x}_{t-k})$ indicates the $k$-th neural network to predict the Granger causal relationship between $\mathbf{x}_{t-k}$ and $\mathbf{x}_t$. The output of $\omega_{\theta_k}(\mathbf{x}_{t-k})$ can be reshaped as a $d \times d$ coefficient matrix, where the entry element $(i, j)$ indicates the influence of $x_{t-k}^{(j)}$ on $x_t^{(i)}$. As shown in Eq. 5, $K$ neural networks are used to predict the weights of past $K$ time legs on deriving $\mathbf{x}_t$. Therefore, relationships between $d$ time series over $K$ time lags can be explored by inspecting $K$ coefficient matrices. Following Eq. 3, we rewrite the Eq. 5 as

$$\mathbf{u}_t = \mathbf{x}_t - \sum_{k=1}^{K} \omega_{\theta_k}(\mathbf{x}_{t-k})\mathbf{x}_{t-k}. \tag{6}$$

Then, given a time series window $\mathbf{W}_t$, we apply the encoder $K$ times to derive the exogenous variables in a window, denoted as $\mathbf{U}_t = (\mathbf{u}_{t-K+1}, \ldots, \mathbf{u}_t)$.

To enforce independence between the derived exogenous variables, we ensure that the distribution of $\mathbf{U}_t$ adheres to an isotropic standard Gaussian distribution $Q$. By assuming that the exogenous variables follow a multivariate Gaussian distribution and applying the KL divergence to quantify the distribution difference, we formulate the independence constraint as

$$\begin{aligned} D_t^{KL}(P(\mathbf{U}_t)\|Q) &= \frac{1}{2}\left(\text{tr}(\Sigma_Q^{-1}\Sigma_t) + (\mu_Q - \mu_t)^T \Sigma_Q^{-1}(\mu_Q - \mu_t) - d + \log\frac{\det\Sigma_Q}{\det\Sigma_t}\right) \\ &= \frac{1}{2}\left(\text{tr}\{\Sigma_t\} + \mu_t^T\mu_t - d - \log\det\Sigma_t\right), \end{aligned} \tag{7}$$

where $\mu_Q = 0$ and $\Sigma_Q = I$ represent the mean and covariance matrix of the isotropic standard Gaussian distribution $Q$; $\mu_t$ and $\Sigma_t$ are the mean and covariance matrix of $\mathbf{U}_t$.

The decoder is to reconstruct the input $\mathbf{x}_t$ based on the exogenous variables $\mathbf{U}_t$. One challenge is that theoretically, the value $\mathbf{x}_t$ at the current time step is computed by the exogenous variables of all the previous time steps. However, considering the potential infinite length of the time series, it is impractical to reconstruct $\mathbf{x}_t$ by using all the previous time steps. To tackle this challenge, we iteratively replace the $\mathbf{x}_{t-k}$ with $\mathbf{x}_{t-(k+1)}$ for a subsequence with length $n$ and derive the following proposition.

**Proposition 1.** *Consider a basic autoregressive model where $\omega_k = \omega_{\theta_k}(\mathbf{x}_{t-k})$ as a framework for analyzing Granger causality. The value at the current time step $\mathbf{x}_t$ can be derived by the exogenous variables from a previous window $[\mathbf{u}_{t-1}, ..., \mathbf{u}_{t-K}]$ and the observed time series from a previous window $[\mathbf{x}_{t-K-1}, ..., \mathbf{x}_{t-2K}]$ with the following equation:*

$$\mathbf{x}_t = \sum_{m=1}^{K} \alpha_{K-m}\mathbf{u}_{t-(K-m)} + \alpha_K \mathbf{x}_{t-K} + \sum_{m=2}^{K+1} \alpha_{K+1-m}\sum_{k=m}^{K}\omega_k\mathbf{x}_{t-k-(K+1-m)}, \tag{8}$$

*where $\omega_k$ indicates the parameter of Granger causality, and $\alpha_n = \sum_{i=1}^{n}\omega_n\alpha_{n-i}$, $1 \leq n \leq K$, is a recursive equation with $\alpha_0 = 1$.*

We provide proof of the proposition in the Appendix A.1. Inspired by Proposition 1, we propose a decoder structure that combines both observed time series and exogenous variables. Specifically, we parameterize the impact of exogenous variable $\mathbf{u}_{t-k}$ on $\mathbf{x}_t$ by a neural network $\bar{\omega}_{\bar{\theta}_k}$ and the impact of observed time series $\mathbf{x}_{t-K-k}$ on $\mathbf{x}_t$ by another neural network $\bar{\omega}'_{\bar{\theta}'_k}$. Then, the decoder computes $\mathbf{x}_t$ based on the following equation.

$$\hat{\mathbf{x}}_t = \sum_{k=1}^{K} \bar{\omega}_{\bar{\theta}_k}(\mathbf{u}_{t-k})\mathbf{u}_{t-k} + \sum_{k=1}^{K} \bar{\omega}'_{\bar{\theta}'_k}(\mathbf{x}_{t-K-k})\mathbf{x}_{t-K-k} + \mathbf{u}_t, \tag{9}$$

where $\hat{\mathbf{x}}_t$ indicates the reconstructed value at time step $t$, and $\mathbf{u}_{t-k}$ is computed by encoder defined in Eq. 6.

The whole encoder-decoder structure can be defined as $\hat{\mathbf{x}}_t = AE_{\theta_k,\bar{\theta}_k,\bar{\theta}'_k}(\mathbf{x}_{<t})$. Given a time series with length $T$, the objective function to train the encoder neural netework $\omega_{\theta_k}$ and decoder neural networks $\bar{\omega}_{\bar{\theta}_k}, \omega'_{\bar{\theta}'_k}$ is defined as:

$$\mathcal{L} = \sum_{t=K+1}^{T} \left\{ \|\hat{\mathbf{x}}_t - \mathbf{x}_t\|_2 + \beta D_t^{KL} + \lambda_{en} R(\Omega_t) + \lambda_{de} R(\bar{\Omega}_t) + \lambda_{de} R(\bar{\Omega}'_t) \right\}$$
$$+ \sum_{t=K+1}^{T-1} \left\{ \gamma_{en} S(\Omega_{t+1}, \Omega_t) + \gamma_{de} S(\bar{\Omega}_{t+1}, \bar{\Omega}_t) + \gamma_{de} S(\bar{\Omega}'_{t+1}, \bar{\Omega}'_t) \right\}, \tag{10}$$

where $D_t^{KL}$ indicates the independence constraint on $\mathbf{U}_t$ defined in Eq. 7; $\Omega_t := [\omega_{\theta_K}(\mathbf{x}_{t-K}) : \cdots : \omega_{\theta_1}(\mathbf{x}_{t-1})]$ indicates the concatenation of coefficient matrices over the past $K$ time steps; similarly, we have $\bar{\Omega}_t := [\bar{\omega}_{\theta_K}(\mathbf{u}_{t-K}) : \cdots : \bar{\omega}_{\theta_1}(\mathbf{u}_{t-1})]$ and $\bar{\Omega}'_t := [\bar{\omega}'_{\theta'_K}(\mathbf{x}_{t-2K}) : \cdots : \bar{\omega}'_{\theta'_1}(\mathbf{x}_{t-K-1})]$; $R(\cdot)$ indicates the L1 and L2 norm penalty for sparsity of the coefficient matrices from the encoder and decoder; the $S(\cdot, \cdot)$ is a smoothness penalty, defined as $S(\Omega_{t+1}, \Omega_t) = \|\Omega_{t+1} - \Omega_t\|_2$; $\lambda$ and $\gamma$ are hyperparameters.

**Granger Causal Discovery**. As the encoder-decoder is proposed to simulate the data generation process governed by Granger causality, we expect the function $\omega_{\theta_k}$ can capture the causal relationships in time series. To further summarize the Granger causal relationships between variables as a summary causal graph, similar to (Marcinkevičs & Vogt, 2021), we aggregate the output from $\omega_{\theta_k}$ into a summarized coefficient matrix as

$$S_{i,j} = \max_{1 \le k \le K} \{\text{median}_{K+1 \le t \le T}(|(\omega_{\theta_k}(\mathbf{x}_{t-k}))_{i,j}|)\}, \text{ for } 1 \le i, j \le d,$$

where $S_{i,j}$ indicates the strength of the Granger causal effect from $\mathbf{x}^{(i)}$ on $\mathbf{x}^{(j)}$. To further derive the adjacency matrix $A$, we set a threshold $\tau$, if the value $S_{i,j} > \tau$, then $A_{i,j} = 1$. In experiments, the threshold is set based on the quantile of the coefficient matrix $S$.

### 4.3 ROOT CAUSE LOCALIZATION

After training on the normal time series, we expect that the exogenous variables can be approximated by the encoder. When deploying the model for root cause localization, we assume the time series is arrived in a streaming manner. When a new time step $t^*$ is arrived, we first adopt the encoder to derive the exogenous variables $\mathbf{u}_{t^*}$ based on Eq. 6. Then, for each time series, $u_{t^*}^{(j)}$, we compute the z-score as the root cause score $z_{t^*}^{(j)} = \frac{u_{t^*}^{(j)} - \mu^{(j)}}{\sigma^{(j)}}$, where $\mu^{(j)}$ and $\sigma^{(j)}$ indicate the mean and standard deviation of the exogenous variable for the $j$-th time series in normal data. We then adopt streaming peaks-over-threshold (SPOT) (Siffer et al., 2017) to dynamically determine the threshold of labeling the potential root cause.

## 5 EXPERIMENTS

### 5.1 EXPERIMENTAL SETUP

**Datasets.** We conduct experiments on four synthetic and two real-world datasets. By using the synthetic datasets, we have the ground truth about the structural causal models as well as the root cause of anomalies. For the real-world datasets, we only have information about root cause variables. Therefore, we use the real-world dataset only to evaluate the root cause identification.

*Synthetic Datasets*: **Linear Dataset** (Marcinkevičs & Vogt, 2021) is dataset with linear interaction dynamics. **Nonlinear Dataset** (Absar et al., 2023) is a nonlinear time series dataset. **Lotka-Volterra**

(Marcinkevičs & Vogt, 2021) is a nonlinear model that simulates a prairie ecosystem with multiple species. **Lorenz 96** (Marcinkevičs & Vogt, 2021) a non-linear time series data. We describe the data generation equation and abnormal behavior injection in Appendix A.2.1.

*Real-world Datasets*: **SWaT** (Mathur & Tippenhauer, 2016) is a dataset collected from a testbed that simulates a real-world water treatment plant. The dataset consists of both normal operations and attack scenarios within the water treatment process. **MSDS** (Multi-Source Distributed System) (Nedelkoski et al., 2020) is developed on an OpenStack testbed. Instances of fault injections are identified as anomalies.

Table 1 shows the statistics of datasets for training and evaluation. The number in the parenthesis indicates the dimensions of multivariate time series. The training set only has the normal time series. In the test set, for each sequence in the synthetic datasets, we conduct one or multiple exogenous interventions randomly to generate the anomalies. The last column shows the average number of variables receiving the exogenous interventions. Note that because Lorenz 96 Marcinkevičs & Vogt (2021) has the most complicated interdependencies between time series, we use more training samples to train AERCA for causal discovery.

Table 1: Statistics of Datasets

| Dataset | Training | Test | | |
|---|---|---|---|---|
| | # of Time Step | # of Sequences ($|\mathcal{X}|$) | Avg. Len. ($\mathbf{T}$) | Avg. # of Root Variables |
| Linear (4) | 5,000 | 100 | 500 | 3.75 |
| Nonlinear (6) | 5,000 | 100 | 500 | 5.25 |
| Lotka-Volterra (40) | 40,000 | 100 | 2,000 | 30.75 |
| Lorenz 96 (20) | 200,000 | 100 | 2,000 | 15.75 |
| SWaT (51) | 49,500 | 20 | 51 | 13.35 |
| MSDS (10) | 29,268 | 4,255 | 21 | 3.05 |

**Evaluation Metrics.** Because root cause identification is achieved based on the learned causal models, we evaluate AERCA in both causal discovery and root cause identification.

*Causal Discovery.* We adopt the commonly used metrics to evaluate the performance of causal discovery (Moraffah et al., 2021; Hasan et al., 2023; Assaad et al., 2022; Sun et al., 2021; Nauta et al., 2019; Pamfil et al., 2020), including F1, AUC-ROC, AUC-PR, and Hamming distance (HD). In the context of causal discovery, F1, AUC-ROC, and AUC-PR quantify the correctness of edge discovery, while Hamming distance calculates the proportion of disagreeing edges between the learned causal graph and the ground truth causal graph.

*Root Cause Identification.* Following the existing work (Ikram et al., 2022; Li et al., 2022; Yu et al., 2021; Ma et al., 2020), we adopt the "recall at top-k" to evaluate the performance of root cause identification, denoted as $AC@K$. This metric quantifies the probability of identifying the correct root cause in the list of variables with the top-k highest root cause scores. Given a set of time series $\mathcal{X}$, the definition of $AC@K$ is shown below.

$$AC@K = \frac{1}{|\mathcal{X}|} \sum_{\mathbf{X} \in \mathcal{X}} \frac{|V_{\mathbf{X}}^{(RC)} \cap \{R_{\mathbf{X}}[k]|k=1,2,...K\}|}{min(K, |V_{\mathbf{X}}^{(RC)}|)},$$

where $R_{\mathbf{X}}[k]$ indicates the time series at the $k$-th rank for the sequence $\mathbf{X}$, and $V_{\mathbf{X}}^{(RC)}$ indicates a set of root cause variables over the whole time series $\mathbf{X}$. Note that if a time series receives multiple exogenous interventions, it only counts as one root cause time series in $V_{\mathbf{X}}^{(RC)}$. We further compute the overall performance by computing the average $AC@K$, denoted as $Avg@K = \frac{1}{K}\sum_{k=1}^{K} AC@k$.

AC@K quantifies the performance of root cause analysis as long as the approach finds the root cause time series. However, in some cases, a time series can receive exogenous interventions for multiple time steps. We also evaluate the effectiveness of approaches for locating the root cause time series at specific time steps. The experimental results are shown in Appendix A.2.3.

The experimental results are reported as the average of five independent runs. Our code is publicly available at https://github.com/hanxiao0607/AERCA.

**Baselines.** We choose two sets of baselines to compare the performance of causal discovery and root cause identification.

*Causal Discovery.* We compare AERCA for the causal discovery with the following baselines. 1) **VAR** (Vector AutoRegressive) (Lütkepohl, 2005) is a linear model to analyze and predict the temporal interdependencies between multiple time series datasets; 2) **cMLP** (Tank et al., 2021) indicates structured multilayer perceptrons (MLPs) combined with sparsity penalties on the weights for Granger causal discovery; 3) **cLSTM** (Tank et al., 2021) leverages recurrent neural networks (RNNs) for Granger causal discovery; 4) **TCDF** (Temporal Causal Discovery Framework) (Nauta et al., 2019) uses attention-based convolutional neural networks for causal discovery from time series data; 5) **eSRU** (economy-Statistical Recurrent Units) (Khanna & Tan, 2019) leverages a special type of RNN called the statistical recurrent unit (SRU) for inferring the Granger causality; 6) **PCMCI** (Runge et al., 2019) combines linear or nonlinear conditional independence tests with a causal discovery algorithm to estimate causal networks; 7) **PCMCI+** (Runge, 2020) extends PCMCI to include discovery of contemporaneous links. 8) **GVAR** (Marcinkevičs & Vogt, 2021) is a vector autoregression with generalized coefficient matrices predicted by neural networks; 9) **CUTS** (Cheng et al., 2023) is a neural Granger causal discovery algorithm building a causal adjacency matrix with imputed data under sparse penalty.

*Root Cause Identification.* We compare AERCA for root cause identification with the following baselines. 1) $\epsilon$-**Diagnosis** (Shan et al., 2019) assumes that the root cause nodes have significantly changed between the abnormal and normal periods and conducts pair-wise significant tests to locate the root cause; 2) **RCD** (Root Cause Discovery) (Ikram et al., 2022) learns the partial causal graph related to the root cause and locate the root cause as the interventional targets; 3) **CIRCA** (Causal Inference-based Root Cause Analysis) (Li et al., 2022) builds structural causal graph via domain knowledge and locates the root cause in anomalies as the nodes with significant distribution changes given its parents. All baselines are implemented by PyRCD (Liu et al., 2023).

## 5.2 EXPERIMENTAL RESULTS

Table 2: Overall performance (mean±std.) of causal discovery.

| Model | Linear | | | | Nonlinear | | | |
|---|---|---|---|---|---|---|---|---|
| | F1 | AUC-PR | AUC-ROC | HD | F1 | AUC-PR | AUC-ROC | HD |
| VAR | $0.969_{\pm0.019}$ | $0.998_{\pm0.003}$ | $0.999_{\pm0.001}$ | $0.011_{\pm0.009}$ | $0.473_{\pm0.164}$ | $0.529_{\pm0.181}$ | $0.676_{\pm0.140}$ | $0.258_{\pm0.130}$ |
| cMLP | $0.745_{\pm0.029}$ | $0.595_{\pm0.038}$ | $0.829_{\pm0.0.25}$ | $0.229_{\pm0.033}$ | $0.419_{\pm0.134}$ | $0.327_{\pm0.079}$ | $0.609_{\pm0.089}$ | $0.340_{\pm0.217}$ |
| cLSTM | $0.684_{\pm0.042}$ | $0.522_{\pm0.048}$ | $0.766_{\pm0.047}$ | $0.312_{\pm0.062}$ | $0.378_{\pm0.000}$ | $0.233_{\pm0.000}$ | $0.500_{\pm0.000}$ | $0.767_{\pm0.000}$ |
| TCDF | $0.943_{\pm0.070}$ | $0.933_{\pm0.081}$ | $0.950_{\pm0.061}$ | $0.033_{\pm0.040}$ | $0.473_{\pm0.107}$ | $0.343_{\pm0.072}$ | $0.655_{\pm0.087}$ | $0.307_{\pm0.065}$ |
| eSRU | $0.964_{\pm0.070}$ | $0.958_{\pm0.082}$ | $0.969_{\pm0.061}$ | $0.021_{\pm0.041}$ | $0.408_{\pm0.152}$ | $0.332_{\pm0.071}$ | $0.617_{\pm0.092}$ | $0.267_{\pm0.069}$ |
| PCMCI | $0.969_{\pm0.031}$ | $0.981_{\pm0.040}$ | $0.986_{\pm0.042}$ | $0.025_{\pm0.038}$ | $0.607_{\pm0.094}$ | $0.456_{\pm0.172}$ | $0.742_{\pm0.147}$ | $0.273_{\pm0.175}$ |
| PCMCI+ | $1.000_{\pm0.000}$ | $1.000_{\pm0.000}$ | $1.000_{\pm0.000}$ | $0.000_{\pm0.000}$ | $0.505_{\pm0.141}$ | $0.410_{\pm0.133}$ | $0.669_{\pm0.134}$ | $0.233_{\pm0.109}$ |
| GVAR | $0.862_{\pm0.052}$ | $0.981_{\pm0.040}$ | $0.986_{\pm0.042}$ | $0.131_{\pm0.066}$ | $0.421_{\pm0.094}$ | $0.562_{\pm0.145}$ | $0.683_{\pm0.097}$ | $0.487_{\pm0.103}$ |
| CUTS | $0.810_{\pm0.076}$ | $0.792_{\pm0.066}$ | $0.844_{\pm0.050}$ | $0.104_{\pm0.034}$ | $0.357_{\pm0.040}$ | $0.249_{\pm0.014}$ | $0.536_{\pm0.032}$ | $0.513_{\pm0.124}$ |
| AERCA | $\mathbf{1.000}_{\pm\mathbf{0.000}}$ | $\mathbf{1.000}_{\pm\mathbf{0.000}}$ | $\mathbf{1.000}_{\pm\mathbf{0.000}}$ | $\mathbf{0.000}_{\pm\mathbf{0.000}}$ | $\mathbf{0.826}_{\pm\mathbf{0.057}}$ | $\mathbf{0.996}_{\pm\mathbf{0.013}}$ | $\mathbf{0.998}_{\pm\mathbf{0.006}}$ | $\mathbf{0.027}_{\pm\mathbf{0.014}}$ |

| Model | Lotka-Volterra | | | | Lorenz 96 | | | |
|---|---|---|---|---|---|---|---|---|
| | F1 | AUC-PR | AUC-ROC | HD | F1 | AUC-PR | AUC-ROC | HD |
| VAR | $0.533_{\pm0.013}$ | $\mathbf{1.000}_{\pm\mathbf{0.000}}$ | $\mathbf{1.000}_{\pm\mathbf{0.000}}$ | $0.044_{\pm0.003}$ | $0.404_{\pm0.162}$ | $0.562_{\pm0.376}$ | $0.764_{\pm0.204}$ | $0.360_{\pm0.121}$ |
| cMLP | $0.511_{\pm0.011}$ | $0.065_{\pm0.014}$ | $0.508_{\pm0.007}$ | $0.049_{\pm0.001}$ | $0.472_{\pm0.058}$ | $0.202_{\pm0.027}$ | $0.569_{\pm0.038}$ | $0.193_{\pm0.031}$ |
| cLSTM | $0.356_{\pm0.176}$ | $0.052_{\pm0.001}$ | $0.500_{\pm0.000}$ | $0.400_{\pm0.428}$ | $0.453_{\pm0.048}$ | $0.194_{\pm0.021}$ | $0.572_{\pm0.004}$ | $0.232_{\pm0.035}$ |
| TCDF | $0.853_{\pm0.032}$ | $0.749_{\pm0.050}$ | $0.890_{\pm0.021}$ | $0.019_{\pm0.002}$ | $0.429_{\pm0.007}$ | $0.290_{\pm0.006}$ | $0.645_{\pm0.004}$ | $0.260_{\pm0.011}$ |
| eSRU | $0.422_{\pm0.039}$ | $0.323_{\pm0.030}$ | $0.634_{\pm0.016}$ | $0.055_{\pm0.002}$ | $0.195_{\pm0.024}$ | $0.225_{\pm0.009}$ | $0.539_{\pm0.009}$ | $0.215_{\pm0.006}$ |
| PCMCI | $0.465_{\pm0.025}$ | $0.291_{\pm0.019}$ | $0.906_{\pm0.017}$ | $0.109_{\pm0.008}$ | $0.368_{\pm0.004}$ | $0.227_{\pm0.007}$ | $0.680_{\pm0.013}$ | $0.540_{\pm0.021}$ |
| PCMCI+ | $0.709_{\pm0.027}$ | $0.651_{\pm0.121}$ | $0.851_{\pm0.082}$ | $0.024_{\pm0.005}$ | $0.502_{\pm0.020}$ | $0.329_{\pm0.022}$ | $0.709_{\pm0.017}$ | $0.163_{\pm0.009}$ |
| GVAR | $0.787_{\pm0.011}$ | $0.988_{\pm0.015}$ | $0.999_{\pm0.002}$ | $0.027_{\pm0.002}$ | $0.568_{\pm0.330}$ | $0.582_{\pm0.361}$ | $0.776_{\pm0.194}$ | $0.142_{\pm0.109}$ |
| CUTS | $\mathbf{0.877}_{0.031}$ | $0.791_{\pm0.047}$ | $0.892_{\pm0.024}$ | $\mathbf{0.011}_{\pm\mathbf{0.002}}$ | $0.341_{\pm0.003}$ | $0.206_{\pm0.002}$ | $0.621_{\pm0.004}$ | $0.404_{\pm0.012}$ |
| AERCA | $0.857_{\pm0.000}$ | $\mathbf{1.000}_{\pm\mathbf{0.000}}$ | $\mathbf{1.000}_{\pm\mathbf{0.000}}$ | $0.026_{\pm0.000}$ | $\mathbf{0.800}_{\pm\mathbf{0.000}}$ | $\mathbf{0.998}_{\pm\mathbf{0.002}}$ | $\mathbf{0.999}_{\pm\mathbf{0.001}}$ | $\mathbf{0.105}_{\pm\mathbf{0.000}}$ |

**Performance of Causal Discovery**. Table 2 shows the results of AERCA and baselines for causal discovery. For baselines, different approaches can achieve good performance on different datasets. For example, VAR can achieve high F1, AUC-PR, AUC-ROC, and low HD on the Linear dataset, but the performance of VAR on other nonlinear datasets is poor, which is expected as VAR is a linear model. Some other advanced approaches, such as TCDF, GVAR, and CUTS, can achieve good performance on the Lotka-Volterra dataset. However, none of the baseline approaches can achieve satisfactory performance on both Nonlinear and Lorenz96 datasets. In contrast, AERCA achieves the perfect performance on the Linear dataset with 1 F1, AUC-PR, AUC-ROC scores, and 0 HD, indicating AERCA can learn the causal graph without any error. For the three more challenging nonlinear datasets, AERCA can achieve high F1, AUC-PR, and AUC-ROC scores, as well as very low HD, showing the capability of AERCA to discover nonlinear causal relationships.

Table 3: Overall performance (mean±std.) of root cause analysis.

| Dataset | Model | AC@1 | AC@3 | AC@5 | AC@10 | Avg@10 |
|---|---|---|---|---|---|---|
| Linear | $\epsilon$-Diagnosis | $0.900_{\pm 0.300}$ | $0.850_{\pm 0.189}$ | $\mathbf{1.000}_{\pm 0.000}$ | $\mathbf{1.000}_{\pm 0.000}$ | $0.950_{\pm 0.043}$ |
| | RCD | $0.500_{\pm 0.500}$ | $0.817_{\pm 0.189}$ | $\mathbf{1.000}_{\pm 0.000}$ | $\mathbf{1.000}_{\pm 0.000}$ | $0.907_{\pm 0.076}$ |
| | CIRCA | $0.600_{\pm 0.490}$ | $0.800_{\pm 0.306}$ | $\mathbf{1.000}_{\pm 0.000}$ | $\mathbf{1.000}_{\pm 0.000}$ | $0.910_{\pm 0.106}$ |
| | AERCA | $\mathbf{1.000}_{\pm 0.000}$ | $\mathbf{1.000}_{\pm 0.000}$ | $\mathbf{1.000}_{\pm 0.000}$ | $\mathbf{1.000}_{\pm 0.000}$ | $\mathbf{1.000}_{\pm 0.000}$ |
| Nonlinear | $\epsilon$-Diagnosis | $0.400_{\pm 0.490}$ | $0.667_{\pm 0.325}$ | $0.880_{\pm 0.165}$ | $\mathbf{1.000}_{\pm 0.000}$ | $0.837_{\pm 0.139}$ |
| | RCD | $0.600_{\pm 0.490}$ | $0.750_{\pm 0.344}$ | $0.880_{\pm 0.165}$ | $\mathbf{1.000}_{\pm 0.000}$ | $0.878_{\pm 0.118}$ |
| | CIRCA | $0.700_{\pm 0.458}$ | $0.717_{\pm 0.395}$ | $0.835_{\pm 0.295}$ | $\mathbf{1.000}_{\pm 0.000}$ | $0.863_{\pm 0.160}$ |
| | AERCA | $\mathbf{1.000}_{\pm 0.000}$ | $\mathbf{1.000}_{\pm 0.000}$ | $\mathbf{1.000}_{\pm 0.000}$ | $\mathbf{1.000}_{\pm 0.000}$ | $\mathbf{1.000}_{\pm 0.000}$ |
| Lotka-Volterra | $\epsilon$-Diagnosis | $0.100_{\pm 0.300}$ | $0.133_{\pm 0.163}$ | $0.138_{\pm 0.149}$ | $0.247_{\pm 0.188}$ | $0.158_{\pm 0.131}$ |
| | RCD | $0.100_{\pm 0.300}$ | $0.133_{\pm 0.163}$ | $0.138_{\pm 0.149}$ | $0.247_{\pm 0.188}$ | $0.158_{\pm 0.131}$ |
| | CIRCA | $0.120_{\pm 0.325}$ | $0.107_{\pm 0.169}$ | $0.120_{\pm 0.150}$ | $0.225_{\pm 0.230}$ | $0.146_{\pm 0.163}$ |
| | AERCA | $\mathbf{1.000}_{\pm 0.000}$ | $\mathbf{1.000}_{\pm 0.000}$ | $\mathbf{1.000}_{\pm 0.000}$ | $\mathbf{1.000}_{\pm 0.000}$ | $\mathbf{1.000}_{\pm 0.000}$ |
| Lorenz96 | $\epsilon$-Diagnosis | $0.100_{\pm 0.300}$ | $0.200_{\pm 0.221}$ | $0.280_{\pm 0.312}$ | $0.450_{\pm 0.330}$ | $0.314_{\pm 0.225}$ |
| | RCD | $0.200_{\pm 0.400}$ | $0.333_{\pm 0.333}$ | $0.400_{\pm 0.358}$ | $0.556_{\pm 0.337}$ | $0.421_{\pm 0.278}$ |
| | CIRCA | $0.360_{\pm 0.480}$ | $0.330_{\pm 0.244}$ | $0.346_{\pm 0.249}$ | $0.539_{\pm 0.263}$ | $0.408_{\pm 0.220}$ |
| | AERCA | $\mathbf{0.996}_{\pm 0.009}$ | $\mathbf{0.996}_{\pm 0.009}$ | $\mathbf{0.997}_{\pm 0.008}$ | $\mathbf{0.996}_{\pm 0.008}$ | $\mathbf{0.990}_{\pm 0.011}$ |
| SWaT | $\epsilon$-Diagnosis | $0.075_{\pm 0.179}$ | $0.125_{\pm 0.217}$ | $0.125_{\pm 0.217}$ | $0.375_{\pm 0.383}$ | $0.180_{\pm 0.194}$ |
| | RCD | $0.000_{\pm 0.000}$ | $0.000_{\pm 0.000}$ | $0.000_{\pm 0.000}$ | $0.300_{\pm 0.458}$ | $0.100_{\pm 0.161}$ |
| | CIRCA | $0.000_{\pm 0.000}$ | $0.000_{\pm 0.000}$ | $0.000_{\pm 0.000}$ | $0.300_{\pm 0.458}$ | $0.100_{\pm 0.161}$ |
| | AERCA | $\mathbf{0.220}_{\pm 0.111}$ | $\mathbf{0.290}_{\pm 0.088}$ | $\mathbf{0.330}_{\pm 0.048}$ | $\mathbf{0.455}_{\pm 0.044}$ | $\mathbf{0.342}_{\pm 0.052}$ |
| MSDS | $\epsilon$-Diagnosis | $0.004_{\pm 0.004}$ | $0.266_{\pm 0.002}$ | $0.452_{\pm 0.009}$ | $\mathbf{1.000}_{\pm 0.000}$ | $0.492_{\pm 0.001}$ |
| | RCD | $0.412_{\pm 0.048}$ | $0.573_{\pm 0.010}$ | $\mathbf{0.984}_{\pm 0.001}$ | $\mathbf{1.000}_{\pm 0.000}$ | $0.821_{\pm 0.012}$ |
| | CIRCA | $\mathbf{0.454}_{\pm 0.238}$ | $0.860_{\pm 0.140}$ | $0.917_{\pm 0.084}$ | $\mathbf{1.000}_{\pm 0.000}$ | $0.809_{\pm 0.035}$ |
| | AERCA | $0.381_{\pm 0.408}$ | $\mathbf{0.908}_{\pm 0.062}$ | $0.974_{\pm 0.027}$ | $\mathbf{1.000}_{\pm 0.000}$ | $\mathbf{0.896}_{\pm 0.037}$ |

**Performance of Root Cause Identification**. Table 3 presents the results of AERCA and baseline models for root cause identification across synthetic and real-world datasets. Although baselines show promising results in terms of AC@5 and AC@10 on both Linear and Nonlinear datasets, it's important to note that these datasets only have a few time series (low dimensionality). In contrast, AERCA shows exceptional performance across all datasets, excluding the SWaT dataset, even at the AC@1 metric, indicating its capability to accurately identify the time series with the highest root cause score. For the SWaT dataset, we observe that the performance of all methods declines, likely due to violations of assumptions such as independence and the presence of complex causal relationships, including hidden confounders and instantaneous effects. However, our AERCA method consistently identifies root cause time series with significantly higher accuracy compared to the baseline methods.

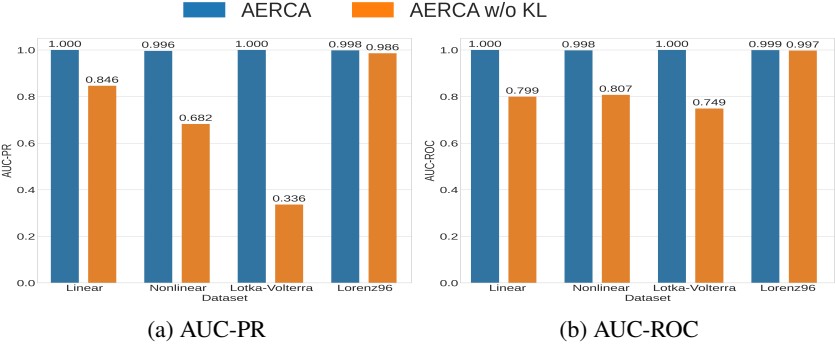

(a) AUC-PR (b) AUC-ROC

Figure 2: Impact of the independent constraint on exogenous variables (Eq. 7) for causal discovery.

**Ablation Study**. To properly learn the exogenous variables, it is critical to ensure the exogenous variables of different time series are independent of each other. Therefore, we have an independent constraint on exogenous variables defined in Eq. 7. To show the importance of the independent constraint for causal discovery, we conduct the ablation study to compare the performance of causal discovery when AERCA is trained with and without the independent constraint in the objective function. As shown in Figure 2, on Linear, Nonlinear, and Lotka-Volterra datasets, without the

independent constraint on exogenous variables, the performance of AERCA for causal discovery is much worse. On Lorenz96, the impact is minor, but without the independent constraint, the performance of AERCA is still lower. The experimental results demonstrate the importance of the independent constraint on exogenous variables for causal discovery.

We show more experimental results in Appendix A.2, including root cause identification at specific time steps, sensitivity analysis on the continuous erogenous interventions, and a case study.

## 6 Conclusions

In this paper, we have developed AERCA for the root cause analysis of anomalies in multivariate time series through Granger causal discovery. AERCA assumes that the anomalies are caused by external interventions on exogenous variables. To achieve root cause analysis, AERCA explicitly considers the exogenous variables when simulating the data generation process. After training on the normal time series data, AERCA can learn the causal relationships among time series as well as derive the exogenous variables. During deployment, exogenous variables that deviate from normal values will be assigned high root cause scores. Experimental results on multiple datasets demonstrate that AERCA achieves state-of-the-art performance in both causal discovery and root cause identification.

## Acknowledgments

This work was supported in part by NSF 1910284, 2142725, and 2103829.

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

## A  APPENDIX

### A.1  APPROACH

**Proposition 2.** *Consider a basic autoregressive model where $\omega_k = \omega_{\theta_k}(\mathbf{x}_{t-k})$ as a framework for analyzing Granger causality. The value at the current time step $\mathbf{x}_t$ can be derived by the exogenous variables from a previous window $[\mathbf{u}_{t-1}, ..., \mathbf{u}_{t-K}]$ and the observed time series from the immediate previous window $[\mathbf{x}_{t-K-1}, ..., \mathbf{x}_{t-2K}]$ with the following equation:*

$$\mathbf{x}_t = \sum_{m=1}^{K} \alpha_{K-m}\mathbf{u}_{t-(K-m)} + \alpha_K\mathbf{x}_{t-K} + \sum_{m=2}^{K+1} \alpha_{K+1-m} \sum_{k=m}^{K} \omega_k\mathbf{x}_{t-k-(K+1-m)}, \quad (11)$$

*where $\omega_k$ indicates the parameter of Granger causality, and $\alpha_n = \sum_{i=1}^{n} \omega_n\alpha_{n-i}, 1 \le n \le K$, is a recursive equation with $\alpha_0 = 1$.*

*Proof.* According to the definition of Granger causality, we have

$$\mathbf{x}_t = \sum_{k=1}^{K} \omega_k\mathbf{x}_{t-k} + \mu_t. \quad (12)$$

Then, we recursively replace variables in the previous window with their autoregressive equations. By substituting $\mathbf{x}_{t-1}$ with $\sum_{k=1}^{K} \omega_k x_{t-1-k} + \mu_{t-1}$, we have

$$\mathbf{x}_t = \omega_1\mathbf{x}_{t-1} + \sum_{k=2}^{K} \omega_k\mathbf{x}_{t-k} + \mu_t$$

$$= \omega_1\Big(\omega_1\mathbf{x}_{t-2} + \sum_{k=2}^{K} \omega_k\mathbf{x}_{t-1-k} + \mu_{t-1}\Big) + \Big(\omega_2\mathbf{x}_{t-2} + \sum_{k=3}^{K} \omega_k\mathbf{x}_{t-k} + \mu_t\Big)$$

By further substituting $\mathbf{x}_{t-2}$, we have

$$\mathbf{x}_t = \big(\omega_1^2 + \omega_2\big)\Big(\omega_1\mathbf{x}_{t-3} + \sum_{k=2}^{K} \omega_k\mathbf{x}_{t-2-k} + \mu_{t-2}\Big) + \omega_1\Big(\omega_2\mathbf{x}_{t-3} + \sum_{k=3}^{K} \omega_k\mathbf{x}_{t-1-k} + \mu_{t-1}\Big)$$

$$+ \Big(\omega_3\mathbf{x}_{t-3} + \sum_{k=4}^{K} \omega_k\mathbf{x}_{t-k} + \mu_t\Big)$$

By defining $\alpha_n = \sum_{i=1}^{n} \omega_n\alpha_{n-i}, 1 \le n \le K, \alpha_0 = 1$, the above expression can be rewritten as

$$\mathbf{x}_t = \alpha_2\Big(\omega_1\mathbf{x}_{t-3} + \sum_{k=2}^{K} \omega_k\mathbf{x}_{t-2-k} + \mu_{t-2}\Big) + \alpha_1\Big(\omega_2\mathbf{x}_{t-3} + \sum_{k=3}^{K} \omega_k\mathbf{x}_{t-1-k} + \mu_{t-1}\Big)$$

$$+ \alpha_0\Big(\omega_3\mathbf{x}_{t-3} + \sum_{k=4}^{K} \omega_k\mathbf{x}_{t-k} + \mu_t\Big)$$

By following this pattern, for any $n$, $1 \leq n \leq K$, we have that

$$\mathbf{x}_t = \alpha_{n-1}\Big(\omega_1\mathbf{x}_{t-n} + \sum_{k=2}^{K}\omega_k\mathbf{x}_{t-k-n+1} + \mu_{t-n+1}\Big) + \alpha_{n-2}\Big(\omega_2\mathbf{x}_{t-n} + \sum_{k=3}^{K}\omega_k\mathbf{x}_{t-k-n+2} + \mu_{t-n+2}\Big)$$

$$+ \cdots + \alpha_0\Big(\omega_n\mathbf{x}_{t-n} + \sum_{k=n+1}^{K}\omega_k\mathbf{x}_{t-k} + \mu_t\Big)$$

By rearranging the above expression, we have that

$$\mathbf{x}_t = \sum_{m=1}^{n}\alpha_{n-m}\mathbf{u}_{t-(n-m)} + \alpha_n\mathbf{x}_{t-n} + \sum_{m=2}^{n+1}\alpha_{n+1-m}\sum_{k=m}^{K}\omega_k\mathbf{x}_{t-k-(n+1-m)}$$

Finally, as we focus on representing $\mathbf{x}_t$ by the exogenous variables from a previous window and the observed time series from the immediate previous window, we assign $n = K$. Then, we have that

$$\mathbf{x}_t = \sum_{m=1}^{K}\alpha_{K-m}\mathbf{u}_{t-(K-m)} + \alpha_K\mathbf{x}_{t-K} + \sum_{m=2}^{K+1}\alpha_{K+1-m}\sum_{k=m}^{K}\omega_k\mathbf{x}_{t-k-(K+1-m)}$$

$\square$

## A.2 EXPERIMENTS

### A.2.1 SYNTHETIC DATASETS

*Linear Dataset* Marcinkevičs & Vogt (2021) is a synthetic time series dataset with linear interaction dynamics. The structural equations are defined as:

$$x_t^{(1)} = a_1 x_{t-1}^{(1)} + u_t^{(1)} + \epsilon_t^{(1)},$$
$$x_t^{(2)} = a_2 x_{t-1}^{(2)} + a_3 x_{t-1}^{(1)} + u_t^{(2)} + \epsilon_t^{(2)},$$
$$x_t^{(3)} = a_4 x_{t-1}^{(3)} + a_5 x_{t-1}^{(2)} + u_t^{(3)} + \epsilon_t^{(3)},$$
$$x_t^{(4)} = a_6 x_{t-1}^{(4)} + a_7 x_{t-1}^{(2)} + a_8 x_{t-1}^{(3)} + u_t^{(4)} + \epsilon_t^{(4)},$$

where coefficients $a_i \sim \mathcal{U}([-0.8, -0.2] \cup [0.2, 0.8])$, additive innovation terms $u_t^{(\cdot)} \sim \mathcal{N}(0, 0.16)$, and anomaly term $\epsilon_t^{(\cdot)}$.

*Nonlinear Dataset* Absar et al. (2023) is a synthetic time series dataset with non-linear interaction dynamics, of which the structural equation is defined as:

$$\mathbf{X}_t = A^T \sum_{m=1}^{t}\beta_m \cos(\mathbf{X}_{t-m} + 1) + \epsilon,$$

where $\beta$ is the regression coefficient, and $\epsilon$ represents standard Gaussian noise. The noise scale is kept below 1 and is proportional to the value of $d$. The non-linear relationship between time series is introduced through the cosine function. The adjacency matrix $A$ of the underlying causal graph is generated using the Erdős–Rényi model Newman (2018).

*Lotka-Volterra* Marcinkevičs & Vogt (2021) is a synthetic time series model that simulates a prairie ecosystem with multiple species. The structural equations are defined as:

$$\frac{d\mathbf{x}^{(i)}}{dt} = \alpha\mathbf{x}^{(i)} - \beta\sum_{j \in Pa(\mathbf{x}^{(i)})}\mathbf{y}^{(j)} - \eta(\mathbf{x}^{(i)})^2, \text{ for } 1 \leq j \leq p,$$

$$\frac{d\mathbf{y}^{(j)}}{dt} = \delta\mathbf{y}^{(j)}\sum_{k \in Pa(\mathbf{y}^{(j)})}\mathbf{x}^{(k)} - \rho\mathbf{y}^{(j)}, \text{ for } 1 \leq j \leq p,$$

$$x_t^{(i)} = x^{(i)} + \epsilon_t^{(i)}, \ y_t^{(j)} = y^{(j)} + \epsilon_t^{(j)}, \text{ for } 1 \leq j \leq p,$$

where $\mathbf{x}^{(i)}$ and $\mathbf{y}^{(j)}$ denote the population sizes of prey and predator, respectively; $\alpha, \beta, \eta, \delta, \rho$ are parameters that decide the strengths of interactions, $Pa(\mathbf{x}^{(i)})$ and $Pa(\mathbf{y}^{(j)})$ correspond the Granger Causality between prey and predators for $\mathbf{x}^{(i)}$ and $\mathbf{y}^{(j)}$ respectively, and $\epsilon_t^{(\cdot)}$ is the abnormal term. We simulate 20 prey species and 20 predator species.

*Lorenz 96* Marcinkevičs & Vogt (2021) a synthetic time series data, where the $i$-th variable is defined by the following nonlinear differential equations:

$$\frac{dx^{(i)}}{dt} = (x^{(i+1)} - x^{(i-2)})x^{(i-1)} - x^{(i)} + F, \text{ for } 1 \le i \le d,$$

where $x^{(0)} := x^{(d)}$, $x^{(-1)} := x^{(d-1)}$, and $x^{(d+1)} := x^1$; and $F$ is a constant controlling the nonlinearity of the data.

*Abnormal behavior injection to the synthetic datasets.* For point anomalies, the anomaly term is single or multiple extreme values for randomly selected time series variables at a specific time step $t$. For example, a point anomaly at time step $t$ can be generated with an abnormal term $\boldsymbol{\epsilon}_t = [0, 2, 4, 0]$, which means the second and third time series have extreme values.

For sequential anomalies, the anomaly terms are function-generated values in a given time range. For instance, setting $\epsilon_{t+i}^{(1)} = 0.1 \times i$, for $0 \le i \le n$, will cause a trend anomaly for time series variable $x^{(1)}$; setting $\epsilon_{t+i}^{(1)} \sim \mathcal{N}(0, 0.16)$, for $0 \le i \le n$, will cause a shapelet anomaly; and setting $\epsilon_{t+i}^{(1)} = (a_1 x_{t+2i-1}^{(1)} + u_{t+2i}^{(1)}) + (a_1 x_{t+2i-2}^{(1)} + u_{t+2i-1}^{(1)}) - (a_1 x_{t+i-1}^{(1)} + u_{t+i}^{(1)})$, for $0 \le i \le n$, will cause a seasonal anomaly.

### A.2.2 IMPLEMENTATION DETAILS

We implement distinct neural network configurations tailored to the complexity of the dataset at hand. Specifically, for synthetic datasets, we employ a two-layer feedforward neural network architecture with a hidden dimension of 50, whereas for real-world datasets, the architecture is expanded to eight layers, each boasting a hidden dimension of 1000. Preprocessing of data is standardized across datasets using a MinMax scaler, with further efficiency measures including downsampling of the SWaT dataset at intervals of every 10 seconds and the MSDS dataset every 5 steps. The training framework is anchored by a learning rate of $1 \times 10^{-6}$, with the Adam optimizer facilitating parameter optimization. Hyperparameters $\beta$, $\lambda$, and $\gamma$ are initially set to 1, ensuring a balanced approach to regularization and loss function adjustment. The maximum training epochs are set to 5000, incorporating an early stopping criterion that halts training if no improvement in loss is observed for 20 consecutive epochs. All experiments were conducted on an Ubuntu 20.04 server equipped with an AMD Ryzen 3960X 24-Core processor at 3.8GHz, dual GeForce RTX 3090 GPUs, and 128 GB of RAM. The implementation uses Python 3.9.7 and PyTorch 1.11.0.

### A.2.3 PERFORMANCE OF ROOT CAUSE ANALYSIS AT SPECIFIC TIME STEPS

To quantify the effectiveness of approaches for locating the root cause time series at specific time steps, we further develop the metric "recall at top-k over all time steps" below.

$$AC^*@K = \frac{1}{|\mathcal{X}|} \sum_{\mathbf{X} \in \mathcal{X}} \frac{|\bigcup_{t \in T} V_{\mathbf{x}_t}^{(RC)} \cap \{R_{\mathbf{X}}^*[k]|k = 1, 2, ..., K\}|}{min(K, |\bigcup_{t \in T} V_{\mathbf{x}_t}^{(RC)}|)},$$

where $V_{\mathbf{x}_t}^{(RC)}$ indicates the set of root cause time series at the $t$-th time step; $R_{\mathbf{X}}^*[k]$ indicates the time series at the $k$-th rank over all time steps. Similarly, we also compute the overall performance by computing the average $AC^*@K$, denoted as $Avg^*@K = \frac{1}{K}\sum_{k=1}^{K} AC^*@k$.

For baselines that cannot identify the root cause at specific time steps, we consider the root cause predicted by the baselines indicating the abnormal time series at the last time step in a sliding window.

As shown in Table 4, for the more challenging metric AC*@K, AERCA can achieve high AC*@1 scores on most datasets except the SWaT dataset, meaning that AERCA can successfully detect the root cause at specific time steps. Furthermore, AERCA achieves near-perfect AC*@10 scores on

most datasets. Considering that there are thousands or even tens of thousands of candidates $(T * d)$ on each dataset when trying to highlight the root cause at specific time steps, the performance of AERCA is promising.

Table 4: Overall performance (mean±std.) of root cause analysis at specific time steps.

| Dataset | Model | AC*@1 | AC*@10 | AC*@100 | AC*@500 | Avg*@500 |
|---|---|---|---|---|---|---|
| Linear | $\epsilon$-Diagnosis | $0.000_{\pm0.000}$ | $0.000_{\pm0.000}$ | $0.000_{\pm0.000}$ | $0.250_{\pm0.316}$ | $0.086_{\pm0.125}$ |
| | RCD | $0.000_{\pm0.000}$ | $0.000_{\pm0.000}$ | $0.000_{\pm0.000}$ | $0.150_{\pm0.300}$ | $0.064_{\pm0.160}$ |
| | CIRCA | $0.000_{\pm0.000}$ | $0.000_{\pm0.000}$ | $0.025_{\pm0.075}$ | $0.233_{\pm0.327}$ | $0.088_{\pm0.144}$ |
| | AERCA | $\mathbf{0.763_{\pm0.137}}$ | $\mathbf{0.990_{\pm0.018}}$ | $\mathbf{1.000_{\pm0.000}}$ | $\mathbf{1.000_{\pm0.000}}$ | $\mathbf{0.998_{\pm0.001}}$ |
| Nonlinear | $\epsilon$-Diagnosis | $0.000_{\pm0.000}$ | $0.000_{\pm0.000}$ | $0.000_{\pm0.000}$ | $0.092_{\pm0.142}$ | $0.056_{\pm0.088}$ |
| | RCD | $0.000_{\pm0.000}$ | $0.000_{\pm0.000}$ | $0.080_{\pm0.240}$ | $0.263_{\pm0.405}$ | $0.116_{\pm0.215}$ |
| | CIRCA | $0.000_{\pm0.000}$ | $0.000_{\pm0.000}$ | $0.017_{\pm0.050}$ | $0.160_{\pm0.182}$ | $0.064_{\pm0.075}$ |
| | AERCA | $\mathbf{0.433_{\pm0.132}}$ | $\mathbf{0.830_{\pm0.094}}$ | $\mathbf{0.994_{\pm0.010}}$ | $\mathbf{0.995_{\pm0.009}}$ | $\mathbf{0.987_{\pm0.094}}$ |
| Lotka-Volterra | $\epsilon$-Diagnosis | $0.000_{\pm0.000}$ | $0.000_{\pm0.000}$ | $0.000_{\pm0.000}$ | $0.000_{\pm0.000}$ | $0.000_{\pm0.000}$ |
| | RCD | $0.000_{\pm0.000}$ | $0.000_{\pm0.000}$ | $0.000_{\pm0.000}$ | $0.000_{\pm0.000}$ | $0.000_{\pm0.000}$ |
| | CIRCA | $0.000_{\pm0.000}$ | $0.000_{\pm0.000}$ | $0.000_{\pm0.000}$ | $0.000_{\pm0.000}$ | $0.000_{\pm0.000}$ |
| | AERCA | $\mathbf{0.997_{\pm0.005}}$ | $\mathbf{0.998_{\pm0.004}}$ | $\mathbf{1.000_{\pm0.000}}$ | $\mathbf{1.000_{\pm0.000}}$ | $\mathbf{1.000_{\pm0.000}}$ |
| Lorenz96 | $\epsilon$-Diagnosis | $0.000_{\pm0.000}$ | $0.000_{\pm0.000}$ | $0.000_{\pm0.000}$ | $0.000_{\pm0.000}$ | $0.000_{\pm0.000}$ |
| | RCD | $0.000_{\pm0.000}$ | $0.000_{\pm0.000}$ | $0.000_{\pm0.000}$ | $0.000_{\pm0.000}$ | $0.000_{\pm0.000}$ |
| | CIRCA | $0.000_{\pm0.000}$ | $0.000_{\pm0.000}$ | $0.000_{\pm0.000}$ | $0.000_{\pm0.000}$ | $0.000_{\pm0.000}$ |
| | AERCA | $\mathbf{0.842_{\pm0.016}}$ | $\mathbf{0.970_{\pm0.013}}$ | $\mathbf{0.996_{\pm0.009}}$ | $\mathbf{0.996_{\pm0.009}}$ | $\mathbf{0.987_{\pm0.010}}$ |
| SWaT | $\epsilon$-Diagnosis | $0.075_{\pm0.179}$ | $0.125_{\pm0.217}$ | $0.125_{\pm0.217}$ | $\mathbf{1.000_{\pm0.000}}$ | $0.633_{\pm0.128}$ |
| | RCD | $0.000_{\pm0.000}$ | $0.025_{\pm0.043}$ | $0.214_{\pm0.066}$ | $0.799_{\pm0.070}$ | $0.416_{\pm0.028}$ |
| | CIRCA | $0.000_{\pm0.000}$ | $0.025_{\pm0.043}$ | $0.214_{\pm0.066}$ | $0.799_{\pm0.070}$ | $0.416_{\pm0.028}$ |
| | AERCA | $0.020_{\pm0.026}$ | $\mathbf{0.320_{\pm0.026}}$ | $\mathbf{1.000_{\pm0.000}}$ | $\mathbf{1.000_{\pm0.000}}$ | $\mathbf{0.950_{\pm0.002}}$ |
| MSDS | $\epsilon$-Diagnosis | $0.000_{\pm0.000}$ | $0.389_{\pm0.410}$ | $0.706_{\pm0.310}$ | $\mathbf{1.000_{\pm0.000}}$ | $0.880_{\pm0.113}$ |
| | RCD | $0.025_{\pm0.026}$ | $0.216_{\pm0.806}$ | $0.806_{\pm0.205}$ | $\mathbf{1.000_{\pm0.000}}$ | $0.908_{\pm0.078}$ |
| | CIRCA | $0.000_{\pm0.000}$ | $0.102_{\pm0.108}$ | $0.741_{\pm0.273}$ | $\mathbf{1.000_{\pm0.000}}$ | $0.884_{\pm0.083}$ |
| | AERCA | $\mathbf{0.230_{\pm0.004}}$ | $\mathbf{1.000_{\pm0.000}}$ | $\mathbf{1.000_{\pm0.000}}$ | $\mathbf{1.000_{\pm0.000}}$ | $\mathbf{0.997_{\pm0.000}}$ |

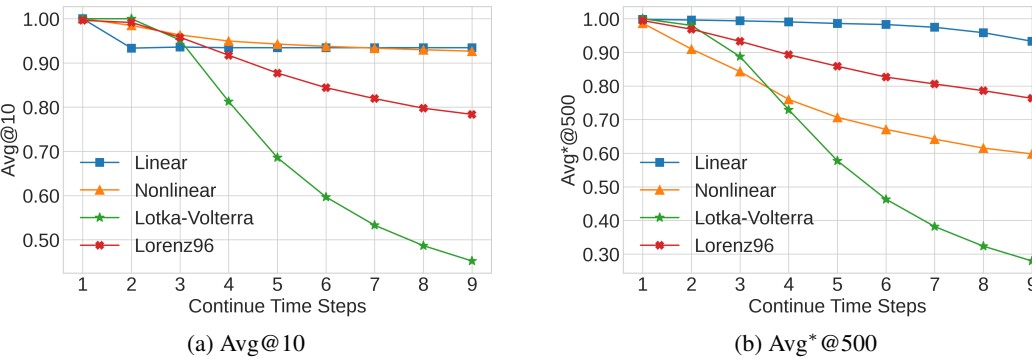

(a) Avg@10

(b) Avg*@500

Figure 3: Performance of root cause identification with various numbers of continuous exogenous interventions.

### A.2.4 SENSITIVITY ANALYSIS

We evaluate the performance of AERCA on root cause identification when the anomalies are caused by continuous exogenous interventions. We tune the number of time steps having the exogenous interventions and check the performance change. Recall that to make the task more challenging, at each time step, the exogenous intervention is conducted on different time series. Figure 3 shows the evaluation results. We can observe that in terms of Avg@10, the performance of AERCA remains stable on both Linear and Nonlinear datasets and slightly decreases on the Lotka-Volterra dataset when increasing the time steps receiving the exogenous interventions. It shows that AERCA can identify the root cause correctly with continuous interventions on different time series. Meanwhile, in terms of Avg*@500, AERCA can still achieve reasonable performance on Linear, Nonlinear,

and Lorenz96 datasets, indicating that in most cases, AERCA can identify the time series receiving exogenous interventions at specific time steps with high root cause scores. The main reason that the performance of AERCA on Lotka-Volterra significantly decreases is because Lotka-Volterra has more variables, which makes the candidates of root cause significantly larger, especially for computing the metric AC*@K. In summary, AERCA can achieve promising performance on root cause identification for continuous interventions when the number of dimensions in multivariate time series is moderate.

### A.2.5  CASE STUDY

Figure 4 shows a short snippet of multivariate time series on the Nonlinear dataset, where we conduct four exogenous interventions on four different time series at different time steps. We highlight the predicted root cause with the top 5 highest root cause scores (AC*@5) via purple bars. We can notice that AERCA correctly detects the root cause time series when exogenous interventions are conducted at specific time steps. Meanwhile, as shown at the bottom of each time series, the distribution of root cause score (z-score) matches the exogenous variables, especially when the time series receives the exogenous interventions.

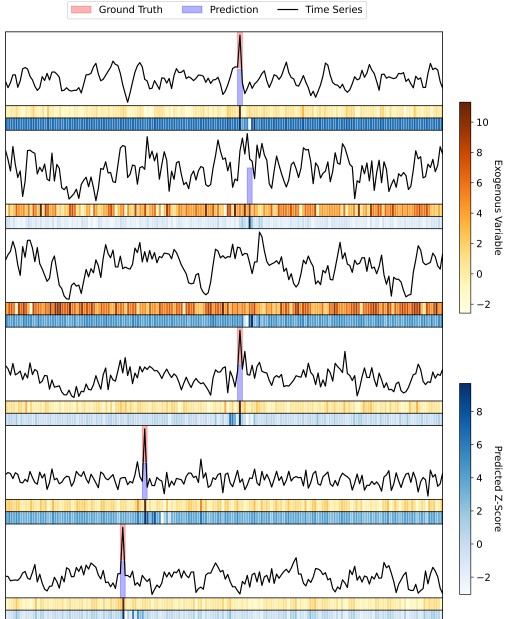

Figure 4: Visualization of multivariate time series, exogenous variables, and predicted root cause scores on the Nonlinear dataset (6 dimensions) with the ground truth and predicted root cause.

