# OpenReview forum: "Root Cause Analysis of Anomalies in Multivariate Time Series through Granger Causal Discovery"
_ICLR.cc/2025/Conference — ICLR 2025 Oral_

### Official Review · Reviewer_sPkL · 2024-10-28

**Soundness:** 4
**Presentation:** 4
**Contribution:** 4
**Rating:** 8
**Confidence:** 4

**Summary:**

This paper proposes AERCA, a novel method for root cause analysis (RCA) based on Granger causality.
AERCA uses an encoder-decoder scheme to iteratively approximate the time series with an approximation of the exogenous variables (noises).
The root causes are modeled as variables that are added to the noises but have significantly higher variance.
By training on normal data (no root causes) AERCA is able to detect the root causes on the test-data.
This was demonstrated empirically on known synthetic (or simulated) and real-world datasets both

**Strengths:**

The paper is very well written and the contribution is clear and novel.
The proposed method is simple and efficient.
In particular, I would like to highlight the ability to handle multiple root causes at once and at multiple time steps, where typically existing work in RCA detects a single root cause.
Moreover, it analytically computes the estimated root causes, rather than backtracking on the computed causal graph (as in existing work in RCA).
I highly appreciate the computational analysis and the result of Proposition 1.
The empirical evaluation is extensive and shows the method's efficiency over prior work in synthetic and real scenarios.

**Weaknesses:**

It is hard for me to point out obvious weakness points. I will however suggest some points for potential improvement, but these do not affect my opinion negatively on the paper.

1. I think it is quite restrictive to assume that normalize data always exist or that we can distinguish them from the faulty ones. How are you sure that the data that you are training on do not contain any root cause (this could affect your training). I am referring to a real scenario where the ground truth is unknown.

2. I would also suggest trying out a bit more intuitive real world experiment where you can show with an example the discovered root cause and its meaning (what was the fault at that node and what did it cause).

3. For the proposed setting of Eq. (5), together with the anomaly term from Eq. (2) are there any theoretical results regarding identifiability? This is well known for some cases ([1] equal variance, [2] imposing sparsity in root causes for static data). Would be interesting to see such a result and it would enhance your contribution.

4. How big time-series can your method handle (how many variables)?

You can discuss the above by adding a limitation section.

[1] Peters, Jonas, and Peter Bühlmann. "Identifiability of Gaussian structural equation models with equal error variances." Biometrika 101.1 (2014): 219-228.
[2] Misiakos, Panagiotis, Chris Wendler, and Markus Püschel. "Learning DAGs from data with few root causes." Advances in Neural Information Processing Systems 36 (2023).

**Questions:**

I would appreciate if you could provide some feedback on the comments on the weaknesses section.

Also, I believe it would be helpful to provide your code as supplementary material for reproduction.

I noted two mistakes/typos:
1. In Eq. (4) you should use $g$ instead of $f$ as this should be a different function depending only on $\mathbf{u}_{\leq t- 1}$
2. In Proposition 1. I would not characterize the window $\mathbf{W}_{t-K}$ as *immediately* previous, rather just previous. Because, before you had sliding windows, each moving by one time-step.

---

> ### Author Response · Authors · 2024-11-21
>
> Thank you for your insightful comments. We will add a limitation section in the revised version. Below are our initial thoughts on these weaknesses.
>
> *W1: In anomaly detection, two common settings are often considered. The first assumes access to normal samples for training, while the second operates under the assumption that the ground truth is unknown. In the latter scenario, it is often typically assumed that the majority of samples are normal, with only a small fraction being anomalous. One future direction could be to assess the performance of our model in cases where the training data contains noise.
>
>
> *W2: Thank you for the suggestion. In our future work, we plan to adopt the Water Network Tool for Resilience (WNTR) testbed to collect normal and abnormal data with ground truth information on root causes.
>
> *W3: We appreciate your suggestion and agree that investigating identifiability in this context is important. We currently don’t have theoretical results on this and plan to explore this in future work.
>
> *W4: The largest number of variables in our experiments is 50 (the SWaT dataset).
>
> *Q2: We released our code on this page (http://tinyurl.com/AERCA2024) for anonymized review and will publish our code in the future.
>
> *Q3: Thank you for the suggestions, and we will revise the paper accordingly.

---

> > ### Comment · Reviewer_sPkL · 2024-11-26
> > **I thank the authors for their clarifications and changes**
> >
> > Thank you for your response and changes in the paper. Most of my concerns are addressed.
> >
> > One thing that I am still concerned with is the scalability. I would consider $50$ variables as a small number for real-world time series. Is this the largest number of variables your method can handle? If yes, what is the bottleneck in your method? Maybe over-parametrization slows it down?

---

> > > ### Author Response · Authors · 2024-11-28
> > >
> > > We agree that scalability is a challenge in causal discovery and is shared across all approaches. However, we think our method achieves comparable or better scalability compared with baselines.
> > > To demonstrate this, we conduct a simple experiment by comparing our method with several baselines on the Lotka-Volterra dataset (40 variables). The results are summarized in the table below:
> > >
> > > | Model    | Time (S) |
> > > |----------|----------|
> > > | VAR      | 159      |
> > > | PCMCI    | 251      |
> > > | PCMCI +  | 974      |
> > > | GVAR     | 1578     |
> > > | AERCA    | 229      |
> > >
> > > We will include the comprehensive results of the runtime comparison with all baselines in the next version of the paper.

---

### Official Review · Reviewer_J7CE · 2024-10-30

**Soundness:** 3
**Presentation:** 3
**Contribution:** 3
**Rating:** 8
**Confidence:** 4

**Summary:**

This paper proposed a method to identify the exogenous variable in multiple time series, such that if the deviation from normal state of this exogenous variable is detected, the cause relation is established. The paper built a Encode-Decode pipeline to implement their design, in which the root cause is defined from Granger causality.

**Strengths:**

Root cause analysis is one of the most challenging problem in multivariate time series analysis. This paper proposed an algorithm to identify the significant deviation of exogenous variable from its normal state, from which one can identify the root cause among multiple time series. Clearly, this is a very interesting approach.

**Weaknesses:**

Due to the assumption that the series are stationary, and the model is based on splitting window with fixed length $K$, thus we may regard the model from neural network as optimization processing in building ARMA model. However, if we regard it as ARMA, the content in (8) of Proposition 1 must be re-checked, since it lacks any criterion of choosing a reasonable window width $K$ (the order of model), and further more, we can not measure the effect of required parameters in the model. Actually, the coefficients of the model in this paper are trained by neural networks, which might be a nonlinear function, thus it might involve serious sensitivity effect.

Secondly, the presentation of this paper is partially confused with standard ARMA model.  I hope the authors could rigrously distinct it with standard ARMA model, and indicates the difference. There actually exist many practical algorithms in time series theory to build an ARMA model as well as detailed performance analysis, which might help to consider the current formulation in more reasonable manner.

In equations (1)(2), the authors lead a description on the potential exogenous intervention on a specific time series at a time step. Then the problem is to detect the abnormal exogenous variable ASAP (Def 1). To my understanding, this exogenous variable is essentially the noise in ARMA model, so the philosophy of this paper is not attractive. That is, I do not agree that the author offers a method to search the root cause which is actually as the change point that affect the trend of time series.

**Questions:**

At line 120-121, the definition of Granger causalitysed in the literature is too sensitive. Could the authors adopt the standard definition, and make the cause effect more statistically significant but not so sensitive?

In Proposition 1, the form of the model is not aggreable with ARMA since in the latter case the width of windows, i.e., the order of model, are determined by some criterions such as AIC. But in current work, the authors didn't give any explaination. Could the authors give some ideas or explanations?

Could the authors describe the statistical property of the exogenrous variable such as distribution, the independence, and the power (variance)?

---

> ### Author Response · Authors · 2024-11-21
>
> *Q1,2,3: Thank you for pointing out this important aspect of our model. The standard definition of Granger causality to which you refer, i.e., $\mathbf{x}\_t=\sum_{k=1}^K w\_k \mathbf{x}\_{t-k} + \mathbf{u}\_t$, is limited to capturing only linear relationships. This limitation is why modern approaches—including ours—leverage deep neural networks to handle non-linearity.
>
> We agree that model order selection is important, and in traditional ARMA models, the order is often determined using criteria such as AIC to balance model complexity and fit. However, this issue may be less significant for our method compared to standard ARMA models because the coefficients in our method are produced by neural networks that take past values of the time series as input. By selecting a sufficiently large window, the neural network can automatically learn which time lags are important, effectively performing model order selection during training. This flexibility allows our method to adaptively capture complex temporal dependencies without relying on predefined model orders.
>
> Our encoder-decoder structure does not assume any distribution for exogenous variables. However, to compute the KL divergence for enforcing independence constraints, we assume that the exogenous variables follow a multivariate Gaussian distribution. Its mean and covariance are estimated from samples during training.

---

> > ### Comment · Reviewer_J7CE · 2024-12-03
> >
> > Thank you so much for your replies.  I've modified my evaluations.

---

### Official Review · Reviewer_QZnY · 2024-11-02

**Soundness:** 3
**Presentation:** 3
**Contribution:** 3
**Rating:** 8
**Confidence:** 3

**Summary:**

This paper introduces AERCA, an autoencoder-based model for root cause analysis in anomaly detection that integrates Granger causal discovery and root cause analysis using an encoder for abductive reasoning to identify exogenous variables and a decoder for deductive reasoning to reconstruct observations. This structure enables AERCA to learn causal relationships and detect anomalies by identifying deviations in exogenous variables.
This approach is evaluated empirically on both synthetic and real-world multivariate time series data. Synthetic datasets include Lotka-Volterra and Lorenz 96, while real-world data comes from SWaT and MSDS.
AERCA outperforms baselines in identifying causal structures and root causes, achieving high F1, AUC-PR, and AUC-ROC scores across four synthetic and two real-world datasets, with strong accuracy in both synthetic and real-world data.

**Strengths:**

This paper has several strengths. It combines Granger causality with anomaly detection, making it suitable for complex systems with multiple interacting variables. This approach identifies not only the affected time series but also the specific time steps, offering precise localization for root cause analysis. The evaluations consider various datasets and baseline models over a range of performance metrics including F1 score and AUC-PR.

**Weaknesses:**

The most noticeable weakness of this approach, in addition to the obvious decreased interpretability due to the reliance on the encoder-decoder autoregressive architecture, is that the model relies strongly on the correct estimation of exogenous variables, which may vary widely in real-world applications.

**Questions:**

Can you describe what is the strategy that will need to be followed by practitioners interested in applying your framework in real world applications?

What assumptions and conditions will an interventional problem need to fulfill for your approach to appropriately model the problem?

---

> ### Author Response · Authors · 2024-11-21
>
> Our method targets the abnormal status that is caused by the exogenous intervention while the underlying causal relationships between time series are unchanged. For instance, consider a cyber-physical system like a water treatment plant equipped with multiple sensors—such as water level, pH level, and electrical conductivity—that generate multivariate time series data. If an attacker overdoses sodium hydroxide, it could lead to abnormal readings in metrics like pH level and electrical conductivity. Our method can be applied to pinpoint the root cause of such abnormal behavior, even when the time series data create ripple effects across other metrics—for example, the increase in sodium hydroxide leading to abnormalities in additional measurements. However, it is important to note that not all attacks can be treated as interventions on exogenous variables, such as attacks on programmable logic controllers (PLCs) that divert water flow. Understanding the nature of anomalies is crucial before applying our method to real-world applications.

---

> > ### Comment · Reviewer_QZnY · 2024-12-03
> >
> > Thank you for the clarification. This is very informative. I have a couple of final suggestions. First, the example you provide is a good motivating case study and could be useful to highlight it briefly in the introduction to motivate the problem. Second, the organizational improvements suggested by other reviewers can be useful in clarifying the methodological contribution of this paper. However, it will be hard for practitioners to adopt the method unless a clear delineation of the application process is indicated. Specifically, I suggest highlighting the model's assumptions (for instance, mutual independence of exogenous variables) in Section 4.

---

> > > ### Author Response · Authors · 2024-12-03
> > >
> > > We appreciate your valuable suggestions and will incorporate these changes in the revised version.

---

### Official Review · Reviewer_ofk8 · 2024-11-02

**Soundness:** 3
**Presentation:** 3
**Contribution:** 3
**Rating:** 8
**Confidence:** 3

**Summary:**

To identify root causes of anomalies in time series data, this paper models anomalies as interventions on exogenous variables. It then proposes a methodology (AERCA) to detect anomalies by explicitly modeling the mutually independent exogenous variables in the underlying autoregressive model, using encoders and decoders. A root cause is then identified by performing statistical tests on each of the associated exogenous variables. Experiments are performed for causal discovery and root cause identification on various synthetic datasets. Root cause identification is also assessed on two real datasets.

**Strengths:**

The problem is well motivated, since root cause identifiaction is prevalent, and existing methods are impractical.

The given definition for root cause, as well as the proposed method are intuitive.

A broad range of system dynamics (ex. linear, nonlinear, lotka-volterra) are considered for empirical evaluation on synthetic datasets.

Writing and overall presentation are quite clear.

**Weaknesses:**

I don't think that this should impact the methodology or results, but the provided definition of Granger causality seems more like actual causality, since the structural equation model is assumed to match the true data-generating process (L112). Although I checked the two references provided, I think that Granger causality is typically understood to be about aiding predictive power given an imperfect model, rather than the true model. Indeed, the causal graph (which is recovered in experiments) is typically defined from the true SEM rather than Granger causality.

The method assumes that the exogenous variables are mutually independent, and this implicitly assumes causal sufficiency (no latent confounders). The importance of this assumption is demonstrated in the ablation experiment. I think that this assumption should be emphasized, since it does not seem like the proposed method is equipped to handle latent confounders.

It is not obvious that the method performs significantly better than alternative methods for root cause identification on real data. On the two offered datasets, performance was poor across all methods for the SwAT dataset, and besides epsilon-diagnosis, results are comprable across methods on the MSDS dataset. This may suggest that there are latent confounders, or other complications in the real datasets, which prevent AERCA from properly modeling the exogenous variables.

Discussion and analysis is limited in the experiments section.

**Questions:**

The importance of mutually independent exogenous variables is demonstrated in the ablation study, such that the method performs significantly worse if the independence constraint is not invoked. Have the authors considered datasets where the exogenous variables are inherently correlated (due to latent confounders)? It would be interesting to see what would happen for root cause identification in this case, since this is a realistic scenario, which would likely be significantly more challenging. It would also be interesting to discuss model misspecification in this case (imposing independence constraint when it is not true).

The method seems to perform poorly on the SWaT dataset. Limited analysis/discussion is offered. Why do the authors think that this is the case? Is it due to higher numbers of root causes, latent confounders, and more complicated causal relationships?

Similarly, is the ground truth causal graph known for the real datasets? It seems likely that the causal graph is not being learnt by AERCA given the performance on root cause identification, which suggests that the exogenous variables are not precisely modeled.

What is the importance of assuming stationarity of the underlying process (L111)?

---

> ### Author Response · Authors · 2024-11-21
>
> *Q1: Thank you for your insightful comment. We have conducted additional experiments by introducing correlations among the exogenous variables in the nonlinear synthetic dataset. The results indicate that such correlations degrade the performance of our method. Addressing hidden confounding will be a valuable direction for future research.
>
> | Dataset       | Type          | F1            | PR            | ROC           | Hamming Dist.  |
> |---------------|---------------|---------------|---------------|---------------|----------------|
> | Nonlinear     | Independent   | 0.826±0.057   | 0.996±0.013   | 0.998±0.008   | 0.027±0.014    |
> | 	          | Dependent     | 0.547±0.169   | 0.795±0.134   | 0.871±0.097   | 0.204±0.265    |
>
> In the dependent case, we set the covariance matrix of exogenous variables as follows:
>
> $
> \text{Covariance Matrix} =
> \begin{bmatrix}
> 1.0 & 0.8 & 0.6 & 0.4 & 0.2 & 0.1 \\\
> 0.8 & 1.0 & 0.7 & 0.5 & 0.3 & 0.2 \\\
> 0.6 & 0.7 & 1.0 & 0.6 & 0.4 & 0.3 \\\
> 0.4 & 0.5 & 0.6 & 1.0 & 0.5 & 0.4 \\\
> 0.2 & 0.3 & 0.4 & 0.5 & 1.0 & 0.6 \\\
> 0.1 & 0.2 & 0.3 & 0.4 & 0.6 & 1.0
> \end{bmatrix}
> $
>
>
>
> *Q2: Thank you for another insightful comment. We agree that the poor performance of all methods (including ours) on the SWaT dataset may be due to violations of the independence assumption or the presence of complex causal relationships. However, since there is no ground truth for this real-world dataset, we cannot verify these hypotheses.
>
>
> *Q3: As mentioned in the above response, we do not have the ground truth causal graph for the real datasets. We agree that the poor performance on SWaT could be due to hidden confounders or more complicated causal relationships.
>
>
> *Q4: The stationary assumption indicates that the underlying causal mechanism is unchanged over time so that we can estimate the causal relationship using neural networks. However, we would like to emphasize that the stationary assumption in our work is largely mitigated compared with the standard Granger causality. This is because our neural networks take past values of the time series as input so that our model can capture time-varying causal relationships and adapt to changes over time to a certain extent.

---

> > ### Comment · Reviewer_ofk8 · 2024-11-25
> > **Response to author's comment**
> >
> > Thank you for clearly addressing my questions and for performing the additional experiment. It makes sense that the method performs significantly worse when the exogenous variables are correlated.
> >
> > Re Q2,3: I see that the lack of a ground truth for these two datasets makes interpretation more challenging. However, I think it would still be good to discuss the possible presence of confounders or more complicated causal relationships when interpreting the results.
> >
> > Re. Granger causality vs SCM discussion: It seems that I am not the only reviewer who was a bit confused by the definition of a Granger causality model via the true structural causal model (do I misunderstand L112?).  For example, reviewer pxU9 asks why Granger causality is used ratehr than SCMs, whereas to me, it seems like in this paper, Granger causality is labeled to be the same as SCM (which is not typical in the literature). I agree with reviewer pxU9 that Granger causality is not commonly regarded to be explicitly causal in this sense. I would advise to clarify this in the paper and recommend either to principally frame it via SCM rather than Granger, which would confuse readers.

---

> > > ### Author Response · Authors · 2024-11-25
> > >
> > > Thanks for your follow-up comments.
> > >
> > > Our definition of Granger causality aligns with the literature (e.g., Eq. (9) in [1], Eq. (4) in [2], Eq. (1) in [3], etc.), where the true causal relationships are represented using structural equations. However, we agree with you and other reviewers that it's important to clarify the distinctions and limitations of Granger causality.
> > >
> > > To address your concerns, we added a corresponding paragraph in Section 3 that highlights these limitations. We also included a discussion of the possible presence of confounders or more complicated causal relationships in the SWaT dataset within the Experimental section. A revised PDF with the modifications highlighted in red has been uploaded.
> > >
> > > [1] Shojaie, Ali, and Emily B. Fox. "Granger causality: A review and recent advances." Annual Review of Statistics and Its Application 9.1 (2022): 289-319.
> > >
> > > [2] Tank, Alex, et al. "Neural granger causality." IEEE Transactions on Pattern Analysis and Machine Intelligence 44.8 (2021): 4267-4279.
> > >
> > > [3] Marcinkevičs, Ričards, and Julia E. Vogt. "Interpretable Models for Granger Causality Using Self-explaining Neural Networks." International Conference on Learning Representations. 2021

---

> > > > ### Comment · Reviewer_ofk8 · 2024-11-27
> > > >
> > > > Thank you for the references. I agree that the definition of Granger causality itself agrees with the referenced literature. Perhaps these should be cited in line 113 for clarity. For further clarity, maybe it would be good to separate the preliminary section as data-generating mechanism/SEM, followed by Granger causality, rather than having Eq. (1) on L107 to be as part of the Granger causality subsection. However, I acknowledge that the current version is basically the same as in [3] Marcinkevičs, Ričards, and Julia E. Vogt. "Interpretable Models for Granger Causality Using Self-explaining Neural Networks." International Conference on Learning Representations. 2021.

---

> > > > > ### Author Response · Authors · 2024-11-27
> > > > >
> > > > > Thank you for your thoughtful feedback. We agree that adding the citations at Line 113 will improve clarity, and we have included them in the revised manuscript.
> > > > >
> > > > > Regarding the structure of the preliminaries section, we appreciate your suggestion to separate it into data-generating mechanisms/SEM and Granger causality. However, we are inclined to follow the convention used in the literature, such as in [1-3], to maintain consistency. We hope this approach remains acceptable.

---

### Official Review · Reviewer_pxU9 · 2024-11-04

**Soundness:** 3
**Presentation:** 2
**Contribution:** 3
**Rating:** 8
**Confidence:** 3

**Summary:**

The paper proposes the AERCA model, an encoder-decoder framework integrating Granger causal discovery with root cause analysis to address anomaly detection in multivariate time series. AERCA identifies causal relationships by defining anomalies as interventions on exogenous variables, capturing deviations that point to root causes. By explicitly modeling the distributions of exogenous variables under normal conditions, AERCA can differentiate abnormal time series data when anomalies arise. The approach achieves promising results on synthetic and real-world datasets, demonstrating its efficacy in both causal discovery and root cause identification.

**Strengths:**

- The integration of Granger causal discovery with root cause analysis via an encoder-decoder structure is a novel approach. This fusion provides a comprehensive method to identify both causal links and root causes of anomalies.
- Modeling exogenous variables under normal conditions adds robustness to anomaly detection, enabling the model to distinguish between normal variations and true anomalies.
- Experiments conducted across multiple datasets, including synthetic and real-world time series, demonstrate AERCA’s high accuracy and adaptability, especially in distinguishing nonlinear causal relationships.
- The model is versatile, as it handles both point and sequence anomalies, making it applicable across various domains requiring real-time anomaly detection.

**Weaknesses:**

- The paper relies on Granger causality to identify causal relationships, which only capture temporal predictability rather than true causation. This can result in spurious relationships, especially in systems with latent confounders or feedback loops, limiting the robustness of causal inference.
- AERCA assumes exogenous variables are mutually independent, which might not hold in real-world scenarios where dependencies can exist between exogenous factors. This assumption could reduce model robustness when applied to systems with interdependencies among exogenous factors.
- While the model structure is powerful, its complexity may hinder interpretability for users, especially when deployed in high-stakes industries like finance or healthcare, where transparent causal explanations are crucial.
- The paper does not extensively discuss the sensitivity of the model to hyperparameters, especially the thresholds for identifying root causes. Further analysis of parameter robustness would help ensure consistent performance across datasets.
- Granger causality is observational and cannot account for unobserved confounders, leading to potential biases in causal inference. Without mechanisms to address these confounders, causal relationships may be inaccurately represented
- One significant limitation of this paper is the omission of several proven and state-of-the-art causal discovery algorithms in its comparison. For example, while PCMCI was used as a baseline, the more efficient PCMCI+ variant (http://auai.org/uai2020/proceedings/579_main_paper.pdf) , which enhances computational performance and accuracy for high-dimensional time series, was not included. Additionally, more recent methods like Kernel-based Causal Discovery (KCD) (https://ieeexplore.ieee.org/iel7/9873850/9873995/09874142.pdf) offer advanced capabilities in handling non-linear dependencies and would have provided a stronger benchmark for evaluation. Conversely, some included approaches, such as Vector Autoregression (VAR) and Temporal Causal Discovery Framework (TCDF), are relatively outdated and come with inherent limitations, particularly in capturing complex non-linear causal relationships. Including these more robust and recent algorithms would enhance the comparative rigor and provide a clearer understanding of the proposed approach’s strengths and weaknesses relative to modern causal discovery methods.

**Questions:**

- Why was Granger causality chosen over more robust causal methods like Structural Causal Models (SCMs) or Dynamic Causal Models (DCMs), given its limitations in capturing true causation?
- How does the model handle scenarios where exogenous variables are interdependent, as this independence assumption may not hold in real-world systems?
- Could the authors provide insights into the model’s sensitivity to key parameter settings, particularly the thresholds for anomaly and root cause detection?
- Given the complexity of the encoder-decoder structure, how do the authors intend to make the model’s reasoning more interpretable for practitioners?
- How does the model address the potential influence of latent confounders, given that Granger causality does not inherently account for unobserved variables?

---

> ### Author Response · Authors · 2024-11-21
>
> *Q1:Granger causality and Structural Causal Models (SCMs) are different but not necessarily conflicting approaches to understanding causal relationships. Traditional Granger causality is limited to pairwise, linear, and stationary causal relationships. Modern Granger causality (including our work) has largely mitigated these limitations by leveraging deep neural networks. To date, combining Granger causality with SCMs, as we did in our paper, is still among the best approaches for discovering temporal causal relationships, since traditional SCMs typically apply to static data. Similar approaches have been widely adopted in the literature (e.g., [1,2]), which inspired our method. No hidden confounding is one of our key assumptions that ensures the reliability of our method. However, we argue that this assumption is reasonable in our context, which will be addressed in our next response.
>
> [1] Pamfil, Roxana, et al. "Dynotears: Structure learning from time-series data." International Conference on Artificial Intelligence and Statistics. Pmlr, 2020.
>
> [2] Marcinkevičs, Ričards, and Julia E. Vogt. "Interpretable Models for Granger Causality Using Self-explaining Neural Networks." International Conference on Learning Representations (ICLR 2021), 2021.
>
> *Q2: As mentioned above, no hidden confounding (i.e., exogenous variables are mutually independent) is one of our key assumptions. Handling scenarios where exogenous variables are interdependent will be a valuable future direction. However, we would like to argue that our method can still be applied reasonably to many practical applications. In cyber-physical systems, depending on how anomalies occur within a causal mechanism, anomalies can be naturally divided into two types: those due to changes in exogenous variables and those due to changes in the structural equations. For example, an attack on a sensor could be treated as the first type, while an attack on a programmable logic controller should be treated as the second type. For the first type of attack, it is relatively reasonable to assume independent exogenous variables, thus supporting the applicability of our method in these cases. However, we agree that relaxing this assumption is important for a wider range of real-world systems and is worthy of further investigation.
>
>
> *Q3: In our approach, the threshold for root cause detection is determined by the approach of streaming peaks-over-threshold (Siffer et al., 2017), which can dynamically determine the threshold. SPOT is based on Extreme Value Theory (EVT), which models the tail of the data distribution to detect the anomalies or root cause nodes, so it is less sensitive to manual hyperparameter tuning.
>
>
>
> *Q4: We agree that the encoder-decoder structure is complex and may lack interpretability. However, we would like to emphasize that our method is more interpretable than standard encoder-decoder structures since it aims to discover the causal relationships in time series. As we mentioned in lines 277-286, the practitioners can construct a summary causal graph from parameters $w_{\theta}$ to gain insights into the model output.
>
> *Q5: Please refer to our response to Q2, as hidden confounding is essentially formulated as dependencies between exogenous variables in SCM.
>
> *Baseline: we compare our approach with a new baseline, PCMCI+. As shown below, our approach achieves better performance than PCMCI+. Regarding another baseline (KCD) suggested by the reviewer, we could not find its source code. However, we will reference this approach in the related work section.
>
> | Dataset        | Model   | F1             | AUC-PR         | AUC-ROC        | HD              |
> |----------------|---------|----------------|----------------|----------------|-----------------|
> | Linear         | PCMCI+  | 1.000±0.000    | 1.000±0.000    | 1.000±0.000    | 0.000±0.000     |
> |                | AERCA   | 1.000±0.000    | 1.000±0.000    | 1.000±0.000    | 0.000±0.000     |
> | Nonlinear      | PCMCI+  | 0.505±0.141    | 0.410±0.133    | 0.669±0.134    | 0.233±0.109     |
> |                | AERCA   | 0.826±0.057    | 0.996±0.013    | 0.998±0.008    | 0.027±0.014     |
> | Lotka-Volterra | PCMCI+  | 0.709±0.027    | 0.651±0.121    | 0.851±0.082    | 0.024±0.005     |
> |                | AERCA   | 0.857±0.000    | 1.000±0.000    | 1.000±0.000    | 0.026±0.000     |
> | Lorenz 96      | PCMCI+  | 0.502±0.020    | 0.329±0.022    | 0.709±0.017    | 0.163±0.009     |
> |                | AERCA   | 0.800±0.000    | 0.998±0.002    | 0.999±0.001    | 0.105±0.000     |

---

> > ### Comment · Reviewer_pxU9 · 2024-11-26
> >
> > Thanks for addressing the concerns and for answering the questions. I particularly appreciate the consideration of the new baseline, PCMCI+ which shows the superiority of AERCA. I respectfully disagree with the argument that Granger causality and SCM are the same (the use of deep neural networks does not necessarily overcome the fundamental assumptions and limitations of Granger Causality). I would recommend authors address the differences between them and the considerations needed to integrate them. You may consider one recent article on Granger causality (Shojaie A, Fox EB. Granger Causality: A Review and Recent Advances. Annu Rev Stat Appl. 2022 Mar;9(1):289-319. doi: 10.1146/annurev-statistics-040120-010930. Epub 2021 Nov 17. PMID: 37840549; PMCID: PMC10571505.) to position your discussion.

---

> > > ### Comment · Reviewer_ofk8 · 2024-11-27
> > > **Agreement about clarifying the distinction between Granger causality and SCM**
> > >
> > > I agree with reviewer pxU9 that the Granger causality notation used in the paper is actually more in line with structural causal models, and could therefore be confusing.

---

> > > ### Author Response · Authors · 2024-11-27
> > >
> > > Thank you for your comment; we agree with your point. We would like to clarify that we do not intend to suggest that Granger causality and SCM are the same. In our response, we mean that modern Granger causality leverages structural equations from SCM to alleviate certain assumptions, such as linearity. In fact, our definitions of Granger causality align with Equation 9 and Definition 1 in [1], the more general notions of Granger causality. However, we agree that it is important to make explicit the limitations of Granger causality. To address a similar question from reviewer okf8, we have included a limitations section in the revised version of the draft. Please let us know whether this addresses your concerns. If not, please suggest any possible revisions that can help avoid potential confusion.
> > >
> > > [1] Shojaie A, Fox EB. Granger Causality: A Review and Recent Advances. Annu Rev Stat Appl. 2022 Mar;9(1):289-319. doi: 10.1146/annurev-statistics-040120-010930. Epub 2021 Nov 17. PMID: 37840549; PMCID: PMC10571505.

---

### Meta-Review · Area_Chair_9HWV · 2024-12-11

**Metareview:**

This paper proposes AERCA, a novel method integrating Granger causal discovery and root cause analysis to identify anomalies in multivariate time series. By modeling anomalies as interventions on exogenous variables and explicitly capturing their normal distributions, the approach provides a robust framework for causal discovery and root cause detection. The methodology is well-supported by experiments on synthetic and real-world datasets, demonstrating strong performance across various evaluation metrics. The paper’s strengths lie in its innovative combination of causal discovery and anomaly detection, effective handling of nonlinear relationships, and adaptability to point and sequence anomalies. While limitations, such as the assumption of independent exogenous variables and sensitivity to latent confounders, were noted, they do not significantly detract from the contribution. The authors addressed key concerns during the rebuttal, further improving the clarity and robustness of the work. Overall, the paper makes a significant contribution to the field of interpretable AI and time-series anomaly detection, warranting a decision of Accept.

**Additional Comments On Reviewer Discussion:**

During the rebuttal period, the authors addressed key concerns raised by reviewers, improving the clarity and robustness of the paper. They clarified the relationship between Granger causality and SCMs, added discussions on the independence assumption of exogenous variables, and conducted experiments showing its impact. Scalability concerns were addressed through additional benchmarks, demonstrating competitive performance. Interpretability was improved by expanding explanations of the encoder-decoder structure, and the inclusion of PCMCI+ as a baseline strengthened the empirical evaluation. These revisions effectively resolved the raised issues, reinforcing confidence in the method’s novelty and practical relevance, supporting a decision to Accept.

---

### Decision · Program_Chairs · 2025-01-22

Accept (Oral)